# GENERATIVE PLANNING FOR TEMPORALLY COORDINATED EXPLORATION IN REINFORCEMENT LEARNING

**Haichao Zhang**    **Wei Xu**    **Haonan Yu**
Horizon Robotics, Cupertino CA 95014
{haichao.zhang, wei.xu, haonan.yu}@horizon.ai

## ABSTRACT

Standard model-free reinforcement learning algorithms optimize a policy that generates the action to be taken in the current time step in order to maximize expected future return. While flexible, it faces difficulties arising from the inefficient exploration due to its single step nature. In this work, we present Generative Planning method (GPM), which can generate actions not only for the current step, but also for a number of future steps (thus termed as *generative planning*). This brings several benefits to GPM. Firstly, since GPM is trained by maximizing value, the plans generated from it can be regarded as intentional action sequences for reaching high value regions. GPM can therefore leverage its generated multi-step plans for temporally coordinated exploration towards high value regions, which is potentially more effective than a sequence of actions generated by perturbing each action at single step level, whose consistent movement decays exponentially with the number of exploration steps. Secondly, starting from a crude initial plan generator, GPM can refine it to be adaptive to the task, which, in return, benefits future explorations. This is potentially more effective than commonly used action-repeat strategy, which is non-adaptive in its form of plans. Additionally, since the multi-step plan can be interpreted as the intent of the agent from now to a span of time period into the future, it offers a more informative and intuitive signal for interpretation. Experiments are conducted on several benchmark environments and the results demonstrated its effectiveness compared with several baseline methods. Code is available: https://github.com/Haichao-Zhang/generative_planning.

## 1 INTRODUCTION

Reinforcement learning (RL) has received great success recently (Mnih et al., 2015; Lillicrap et al., 2016; Gu et al., 2017; Haarnoja et al., 2018), reaching or even surpassing human-level performances on some tasks (*e.g.* Mnih et al., 2015; Silver et al., 2016). However, the number of samples required for training is still quite large, rendering sample efficiency improvement a critical problem. Standard model-free RL methods operate at the granularity of individual time step. Although flexible, one prominent issue is inefficient exploration, as the consecutive one-step actions are not necessarily temporally consistent, which makes the probability of consistent movement decays exponentially with the exploration steps, rendering many of the exploration efforts wasted (Dabney et al., 2021).

Extensive efforts have been made on improving the temporal consistency of exploration from different perspectives with different approaches, including parameter space perturbation (*e.g.* Plappert et al., 2018b; Fortunato et al., 2018; Xie et al., 2019), enhancing correlation between policy steps (*e.g.* Lillicrap et al., 2016; Korenkevych et al., 2019; Amin et al., 2021; Zhang & Van Hoof, 2021) and skill/option-based exploration (*e.g.* Agarwal et al., 2018; Tirumala et al., 2020; Pertsch et al., 2020). Among them, simple action repeat-based temporal persistency has been shown to be very successful (*e.g.* Mnih et al., 2015; Sharma et al., 2017). Action repeat refers to the strategy of repeating a previous action for a fixed or variable number of time steps. This simple strategy demonstrates strong performance on a broad range of tasks and has been actively investigated recently (Neunert et al., 2019; Dabney et al., 2021; Biedenkapp et al., 2021; Yu et al., 2021).

Planning is another effective approach for obtaining a control policy alternative to RL (*e.g.* Chua et al., 2018; Srinivas et al., 2018; Hafner et al., 2019; Eysenbach et al., 2019). Typically, it is used to

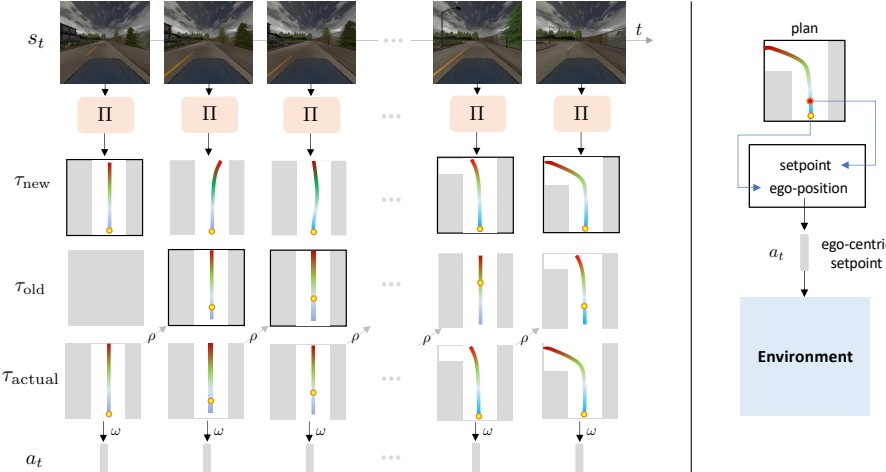

Figure 1: **Generative Planning Method**. GPM has three main components. i) **Plan generation.** At each time step, the plan generator $\Pi$ generates a plan $\tau_{\text{new}}$ based on the state $s_t$. ii) **Plan update.** $\tau_{\text{old}}$ is obtained by "time-forwarding" the previous actual plan $\rho(\tau'_{\text{actual}})$ one step. iii) **Plan switch.** The replanning signal is generated based on the values of the previous plan $\tau_{\text{old}}$ and the new one $\tau_{\text{new}}$, and only switches if certain criteria is satisfied. The plan enclosed with a black box indicates the one been selected at that time step. In the case of driving, the primitive action to be executed is derived from the plan as $\omega(\tau_{\text{actual}})$, which is sent to the environment for execution as shown on the right.

optimize a sequence of actions by online minimization of a cost function computed together with a model (Chua et al., 2018; Srinivas et al., 2018; Hafner et al., 2019). The control loop is closed with model-predictive control (MPC), *i.e.* by executing the first action only and re-run the optimization again in the next time step. Planning excels at reasoning over long horizons and has a high sample efficiency, but is more computationally demanding and requires access to a model (Levine & Koltun, 2013a;b; Lowrey et al., 2019; Srinivas et al., 2018; Chua et al., 2018; Hafner et al., 2019). A large body of work in the literature has explored how to leverage the notable complementary strengths of RL and planning from many perspectives, *e.g.*, guiding policy learning with planning (Levine & Koltun, 2013a;b), improving planning with a learned policy (Schrittwieser et al., 2019; Wang & Ba, 2020) and generating intrinsic reward for exploration (Sekar et al., 2020).

This work is motivated by recognizing both the apparent *complementary strengths* and less apparent *connections* between these two streams of seemingly different approaches, *i.e.*, simple but successful action repeat strategy and flexible but more demanding planning-based control. We note that they can be connected from the perspective of *planning for exploration*. On the one hand, we can go beyond simply repeating the same actions by using planning to induce more flexible forms of action sequences for temporally coordinated exploration. On the other hand, the empirical success of action repeat suggests that the form of planning that is useful for exploration could be possibly done in a way that is lightweight and retains the overall model-free nature of the whole algorithm.

Motivated by these observations, we propose a Generative Planing Method (GPM) as an attempt towards the goal of model-free RL with temporally coordinated, planning-like behavior for exploration. The overall framework is shown in Figure 1. At the beginning of an episode, a plan $\tau$ is generated. The plan is a temporally extended sequence of actions and the one-step action for the current time step of execution is derived from it as $\omega(\tau)$. Upon entering the next time step, the old plan is adjusted by "time-forwarding" one step for alignment as $\rho(\tau)$. The old plan is committed to until certain conditions are met when compared to a new one generated at each time step. By doing this, we have enabled both flexible intentional action sequences and temporally extended exploration. We refer this type of temporally extended and intentional exploration as *temporally coordinated exploration*.

The contributions of this work can be summarized as follows:

- we provide a natural perspective on extending standard model-free RL algorithms to generate multi-step action sequences for intentional exploration via planning by leveraging a connection between planning and temporally extended exploration.

- we propose a generative planning method as a concrete instantiation and initial attempt in this direction;

- we verify the effectiveness of the proposed method by comparing it with a number of baseline methods;

- we show that the proposed method has additionally improved interpretability on several aspects, including both the evolvement of its exploration behavior and final policy behavior.

## 2 BACKGROUND

We will focus on continuous control tasks in this work. We first review some basics on model-free RL for continuous control, temporally extended exploration and model-based planning.

### 2.1 MODEL-FREE RL FOR CONTINUOUS CONTROL

Standard RL methods like SAC (Haarnoja et al., 2018) learn a policy function which maps the current state $s$ to action $a$ as $\pi(s) \to a$. The policy is typically modelled with a neural network with learnable parameters $\theta$ and the optimization of the policy parameters is achieved by maximizing the following function *w.r.t.* $\theta$:

$$\pi_\theta = \arg \max_\theta \mathbb{E}_{s \sim \mathcal{D}} \mathbb{E}_{a \sim \pi_\theta} [Q(s, a) - \alpha \log \pi_\theta(a|s)]. \tag{1}$$

where $\mathcal{D}$ denotes replay buffer and $\alpha$ is a temperature parameter. $Q(s, a)$ denotes the soft state-action value function and is defined as:

$$Q(s, a) = \bar{r}(s, a) + \gamma \mathbb{E}_{s' \sim T(s,a), a' \sim \pi_\theta(s')} [Q(s', a')],$$

where $T(s, a)$ denotes the dynamics function and $\bar{r}(s, a)$ the policy entropy augmented reward function (Haarnoja et al., 2018). $\gamma \in (0, 1)$ is a discount factor. By definition, $Q(s, a)$ represents the accumulated discounted future entropy augmented reward starting from $s$, taking action $a$, and then following policy $\pi_\theta$ thereafter.

### 2.2 TEMPORALLY EXTENDED EXPLORATION MEETS MODEL-BASED PLANNING

The policy of the form in Eqn.(1) generates a one-step action at a time and the exploration is typically achieved by incorporating noise at single step level (Haarnoja et al., 2018). While this is simple and flexible in representing the control policy, it might not be effective during exploration as the consecutive one-step actions are not necessarily temporally consistent, limiting its ability to escape local optima thus wasting many of the exploration efforts (Dabney et al., 2021).

Instead, a directed form of exploration can potentially mitigate this issue by generating consistent actions for a certain period of time (Sutton et al., 1999; Mnih et al., 2015; Sharma et al., 2017; Dabney et al., 2021). Build upon this philosophy, action repeat-based temporally extended exploration, despite of its simplicity, has been proven to be very effective and has been actively explored recently (Sharma et al., 2017; Dabney et al., 2021; Biedenkapp et al., 2021; Yu et al., 2021).

Apart from directly improving the temporal property of the exploration behavior, another set of approaches guide policy optimization with model-based planning (*e.g.* Levine & Koltun, 2013a;b; Lowrey et al., 2019; Hoppe et al., 2019). In model-based planning, the optimized action for a particular time step is obtained by online minimization of a cost function. To better leverage the temporal structure of the cost function, planning is typically done for a span of time period by optimizing a sequence of actions $\tau = [a_0, a_1, \cdots]$ (*e.g.* Williams et al., 2017; Srinivas et al., 2018; Chua et al., 2018; Hafner et al., 2019; Schrittwieser et al., 2019):

$$[a_0, a_1, \cdots] = \arg \min_{\tau = [a_0, a_1, \cdots]} C(s, [a_0, a_1, \cdots]). \tag{2}$$

This type of approaches has the potential to generate more intentional and flexible action sequences than repeating actions, as they are optimized at the sequence level. The generated action sequences is typically used with MPC by executing only the first action and discarding the rest. Also, the prerequisite of an additional model for calculating $C$ and the online optimization of it indicate a large gap between this type of approaches and model-free RL. This makes the task of how to leverage planning for exploration in model-free RL more challenging.

## 3 GENERATIVE PLANNING METHOD FOR MODEL-FREE RL

Model-based planning can generate flexible plans, but requires online planning with a model which is computationally intensive. On the other hand, action-repeat is more efficient, but is less adaptive in its form of plans. This motivates us to present GPM, which supports flexible plans beyond repeating for exploration without online optimization and resembles typical model-free RL algorithms.

We define a plan $\tau$ as a sequence of decisions that spans a number of time steps in the corresponding space $\mathcal{A}$. For example, if $\mathcal{A}$ is the space of primitive actions, $\tau$ is a sequence of primitive actions, *i.e.* $\tau = [a_0, a_1, \cdots]$. If $\mathcal{A}$ is the spatial coordinate space as common in robotics control and driving, then $\tau$ is a sequence of spatial positions and is essentially a *spatial-temporal* trajectory.

The framework of GPM is illustrated in Figure 1. At the core of GPM is a *plan generator* $\Pi_\theta(s) \mapsto \tau$, which is a generative model that takes state $s$ as input and outputs a plan $\tau$ (*c.f.* Section 3.1). In essence, it is a form of policy that outputs multi-step actions. (therefore termed as *generative planning*). The plan generator is trained by maximizing plan values, which shapes it to generate plans that are more adaptive along the training process (*c.f.* Section 3.2). During rollout, at each time step $t$, give the observation $s_t$, the plan generator $\Pi$ generates a plan $\tau_{\text{new}} = [a_t, a_{t+1}, a_{t+2}, \cdots]$. Also, the old plan (if available) from the previous step is "time-forwarded" one step to generate $\tau_{\text{old}} = \rho(\tau'_{\text{actual}})$, where $\tau'_{\text{actual}}$ denotes the actual plan from the previous time step. The replanning mechanism is based on the values of both plans $\tau_{\text{old}}$ and $\tau_{\text{new}}$ to determine which one is used as the actual plan $\tau_{\text{actual}}$ (*c.f.* Section 3.3).

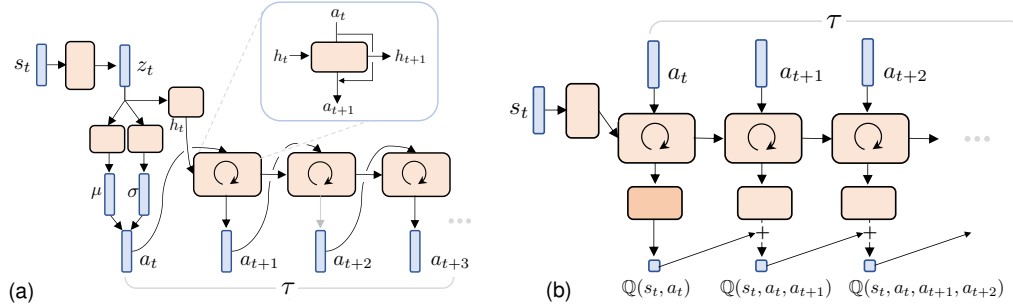

Figure 2: **Network Structures** of (a) plan generator (generative actor) and (b) plan value function (critic). (a) The plan generator generates a sequence of actions in an autoregressive way to form the full plan. For each step start from the second one, a skip-connection is added from the input previous action to the output action. (b) The plan value function takes state and plan as input, and outputs a sequence of values along the plan. In this example, the plan value function takes $s_t$ and $\tau = [a_t, a_{t+1}, a_{t+2}, \cdots]$ as input and outputs values $[\mathbb{Q}(s_t, a_t), \mathbb{Q}(s_t, a_t, a_{t+1}), \mathbb{Q}(s_t, a_t, a_{t+1}, a_{t+2}), \cdots]$. $\mathbb{Q}(s_t, a_t)$ uses a separate decoder different from the rest of steps $t+1, t+2, \cdots$, whose values are computed as the sum of the value from the previous step and the output from the decoder.

### 3.1 PLAN GENERATOR MODELING

It is natural to model the plan generator $\Pi$ with an autoregressive structure to capture the inherent temporal relations between consecutive actions. In practice, we use a recurrent neural network (RNN) to model $\Pi$, with its structure shown in Figure 2 (a). More concretely, the plan generator $\Pi_\theta(s)$ is implemented as a stochastic recurrent network as follows:

$$z_t = \text{encoder}(s_t), \quad h_t = h(z_t)$$
$$a_t = \mu_{\text{mean}}(z_t) + \sigma_{\text{std}}(z_t) \cdot n \qquad n \sim \mathcal{N}(0, 1) \tag{3}$$
$$a_{t+i} = f(h_{t+i}, a_{t+i-1}), \quad h_{t+i} = \text{RNN}(h_{t+i-1}, a_{t+i-1}) \quad i \in [1, n-1].$$

$z$ denotes the feature extracted from $s$. $h(z)$ denotes a network for initializing the state for RNN. In the special case when $\tau$ only contains a single action, Eqn.(3) reduces to the standard stochastic actor as used in Haarnoja et al. (2018), with $\mu_{\text{mean}}$ and $\sigma_{\text{std}}$ the network for predicting mean and standard derivation of the action distribution for the current time step. $f$ denotes the action decoder for future time steps, which can be implemented with some prior incorporated when available. For example, we can implement $f$ with a skip-connection to leverage the temporal structure:

$$a_{t+i} = f(h_{t+i}, a_{t+i-1}) = a_{t+i-1} + g(h_{t+i}, a_{t+i-1}) \qquad i \in [1, n-1] \tag{4}$$

where $g$ is implemented with neural networks. In this case, if we initialize $g$ to generate small values, then we actually incorporated an action repeat prior to start with for exploration and learning. The

degrees of stochastic of $\Pi_\theta(s)$ can be adjusted by matching a specified a target entropy in a way similar to SAC (Haarnoja et al., 2018), contributing a term $J(\alpha)$ into the total cost (*c.f.* Appendix A.1).

## 3.2 Plan Value Learning and Plan Generator Training

We propose to optimize the plan generator by solving the following optimization problem:

$$\Pi_\theta \leftarrow \arg\min_\theta J_\Pi(\theta) = \arg\min_\theta \mathbb{E}_{s\sim\mathcal{D}}\mathbb{E}_{\tau\sim\Pi_\theta, l\sim U(1,L)}[-\mathbb{Q}(s,\tau_l)] \tag{5}$$

where $\tau_l$ denotes sub-plan of length $l$ extracted from $\tau$, starting from its beginning. $U(1,L)$ denotes a discrete uniform distribution and $L$ denotes the maximum plan length. By taking expectation *w.r.t.* $l \sim U(1,L)$, we essentially train the plan generator by maximizing not only the value of the full plan, but also the value of the sub-plans. The *plan value* $\mathbb{Q}(s_t,\tau_l)$ is defined as:

$$\mathbb{Q}(s_t,\tau_l) = \mathbb{Q}(s_t,a_t,a_{t+1},\cdots,a_{t+l-1}) = r_{t+1} + \gamma r_{t+2} + \cdots + \gamma^l \mathbb{E}_{s'\sim T(s,\tau_l),\tau'\sim\pi_\theta(s')}[\mathbb{Q}(s',\tau')].$$

Here $s_t$ denotes the current state, $r_{t+1} = R(s_t,a_t)$, $r_{t+2} = \mathbb{E}_{s_{t+1}\sim P(s_t,a_t)}R(s_{t+1},a_{t+1})$, with $R$ the reward function. $P(s_t,a_t)$ and $T(s,\tau)$ denote the action-level and plan-level transition functions respectively. This formulation has a number of potential benefits: *i*) it can generate flexible and adaptive form of plans to be used for temporally extended exploration; *ii*) the *generative* approach used for plan generation retains the resemblance to typical model-free RL algorithms while avoids expensive online planning; *iii*) it gains additional interpretability because of its multi-step plan.

To train the plan generator, we use the re-parametrization trick as in SAC (Kingma & Welling, 2014; Schulman et al., 2015; Haarnoja et al., 2018), and also leverage the differentiability of the plan generator for gradient calculation: $\Delta\theta = \mathbb{E}_{(s,\tau)\sim\mathcal{D},l\sim U(1,L)}\left[\frac{d\tau_l}{d\theta}\frac{d\mathbb{Q}(s,\tau_l)}{d\tau_l}\right]$. $\mathbb{Q}$ needs to be learned in order to calculate $\Delta\theta$. In practice, we parametrize it with neural networks as $\mathbb{Q}_\phi$, whose structure shown in Figure 2(b). More concretely, the plan value function is implemented as a recurrent network, taking the one-step action as input at each step, and outputting the plan value up to the current step. To leverage the temporal structure, we compute the plan value as the sum of the plan value from the previous step and the output from the decoder of the current step. Since there is no previous step value for the first step of plan, the decoder of the first step is not shared with the rest of plan steps. To train the plan value function $\mathbb{Q}_\phi$, we use TD-learning as commonly used for value learning in RL algorithms (*e.g.* Haarnoja et al., 2018; Fujimoto et al., 2018)

$$\mathbb{Q}_\phi \leftarrow \arg\min_\phi J_\mathbb{Q}(\phi) = \arg\min_\phi \mathbb{E}_{(s,\tau)\sim\mathcal{D},l\sim U(1,L)}\|\mathbb{Q}_\phi(s,\tau_l) - \mathbb{T}(s,\tau_l,\mathbf{r})\|_2^2 \tag{6}$$

where $\mathbb{T}(s,\tau_l,\mathbf{r}) = r_{t+1} + \gamma r_{t+2} + \cdots + \gamma^l \mathbb{E}_{s'\sim T(s,\tau_l),\tau'\sim\pi_\theta(s')}[\mathbb{Q}_{\phi'}(s',\tau')]$ denotes the target value computed with a target critic network $\mathbb{Q}_{\phi'}$, with $\phi'$ denoting the parameter values periodically copied from $\phi$ softly (Haarnoja et al., 2018). By taking expectation *w.r.t.* $l \sim U(1,L)$, we learn not only the value of the full plan, but also the value of the sub-plans, which is used in replanning. In addition, an entropy term (computed on the first step of the plan generator) is also added to the target value following Haarnoja et al. (2018). This form of learning has the following potential benefits: *i*) it can be used to train the plan generator as in Eqn.(5); *ii*) it can learn from trajectories sampled from a replay buffer, which is not necessarily aligned with the plans used during rollout (*c.f.* Appendix A.6.1); *iii*) it supports the evaluation of sub-plans, which is necessary for adaptive replanning, as detailed below.

## 3.3 Temporal Plan Update and Replanning

To encourage temporally coordinated exploration, we propose to commit to the current plan by default and switch if a new plan is better according to certain criteria. More concretely, at the current step, we have a plan candidate $\tau_{\text{old}}$ obtained from the plan $\tau'_{\text{actual}}$ of the previous step as $\tau_{\text{old}} = \rho(\tau'_{\text{actual}})$. In the typical case where the plan is consisted of a sequence of actions, The action to be executed at the current step is $\omega(\tau) = \tau[0]$, and $\rho$ is simply a shifting operator extracting the rest of the plan: $\rho(\tau) = \tau[1:]$, serving as a plan candidate for the current time step. At the same time, we generate another new plan $\tau_{\text{new}} \sim \Pi(s_t)$. Now we can make a decision on which of the two candidates we shall use. If the length of $\tau_{\text{old}}$ (denoted as $l_{\text{old}} = |\tau_{\text{old}}|$) is zero, meaning the previous plan has been fully executed, then we just need to switch to $\tau_{\text{new}}$. Otherwise, we compare their plan values for determining a switch. Note that since $\tau_{\text{old}}$ is always shorter than $\tau_{\text{new}}$, the plan values need to be computed up to $l_{\text{old}}$. Denoting the plan values under this setting as $\mathbb{Q}_{l_{\text{old}}}(s,\tau_{\text{old}})$ and $\mathbb{Q}_{l_{\text{old}}}(s,\tau_{\text{new}})$, we then construct a categorical distribution for generating replanning signals as $\text{replan} \sim \text{Categorical}(\mathbf{z})$, where $\mathbf{z}$ denotes the logits and is defined as $\mathbf{z} \triangleq [\epsilon, \mathbb{Q}_{l_{\text{old}}}(s,\tau_{\text{new}}) - \mathbb{Q}_{l_{\text{old}}}(s,\tau_{\text{old}})]$. If replan=1, we switch to $\tau_{\text{new}}$; otherwise, we commit to

$\tau_{\text{old}}$ at the current time step. Intuitively, if the value of the new plan is much higher than that of the old one, a plan switch is preferred. Otherwise, it prefers to commit to the old plan. To adjust $\epsilon$, we use an objective similar to the entropy-based temperature adjustment in SAC (Haarnoja et al., 2018):

$$J(\epsilon) = \epsilon \cdot (l_{\text{commit}} - l_{\text{commit\_target}}), \tag{7}$$

where $\epsilon \geq 0$, $l_{\text{commit\_target}}$ is a target commit length and $l_{\text{commit}}$ is the actual commitment length computed as the exponential moving average of commitment length during rollout. By doing this, we can better leverage the temporally coordinated plans from the plan generator for exploration, while allowing to switch plans if necessary. The overall algorithm is summarized in Appendix A.1.

## 4 RELATED WORK

**Exploration with Temporally Correlated Action Sequences.** Our work mainly lies in this category. Methods in this category focus on improving the correlations of consecutive actions to improve over simple but widely used exploration methods like $\epsilon$-greedy (Dabney et al., 2021). Some methods focus on making the stochastic process underlying the exploration policy more temporally correlated (*e.g.* Lillicrap et al., 2016; Korenkevych et al., 2019). Action repeat strategy leads to a family of successful methods in this category by fully discarding action space randomness for the repeating steps after the first one (*e.g.* Mnih et al., 2015; Sharma et al., 2017; Neunert et al., 2019; Dabney et al., 2021; Biedenkapp et al., 2021; Yu et al., 2021). The proposed method is connected with action repeat in the sense that the source of stochasticity is also introduced in the first step of the action sequence, but it supports more flexible form of action sequences beyond repeating, and achieves this through plan generator training, rather than relying on certain type of specifically designed stochastic process as in Lillicrap et al. (2016); Korenkevych et al. (2019). This category of methods mostly retain the flat RL structure, in contrast to the more complicated hierarchical RL structures as reviewed below.

**Hierarchical RL.** There is a large body of work on improving exploration and learning efficiency by operating at skill level with a higher level meta-controller (*e.g.* Dayan & Hinton, 1993; Sutton et al., 1999; Randlov, 1999; Vezhnevets et al., 2016; Bacon et al., 2017; Florensa et al., 2017; Co-Reyes et al., 2018; Konidaris et al., 2010; Pertsch et al., 2020). However, most hierarchical RL methods are nontrivial to adapt because of some task-specific aspects, *e.g.* the proper design of skill space, its form of representation and its way of acquisition (Dayan & Hinton, 1993; Sutton et al., 1999; Florensa et al., 2017; Co-Reyes et al., 2018; Konidaris et al., 2010; Pertsch et al., 2020). In contrast, the proposed method is a light-weight design that resembles standard model-free RL algorithms to a great extend, thus lies in a space that is mostly complementary to hierarchical RL methods.

**Goal-conditional RL.** Goal conditional RL methods learn a goal-conditional policy and address the efficiency of policy learning by using techniques such hindsight experience replay. There are some recent efforts on generating goals beyond simply relabeling the experienced states (*e.g.* Ren et al., 2019; Pong et al., 2020; Pitis et al., 2020). In this setting, it resembles the hierarchical RL structure, where the goal generator plays the role of a high-level meta-controller and the goal-conditional policy functions as a low-level worker. Differently, we focus on how to improve action sequence based exploration for plain model-free RL algorithms without resorting to goal-oriented designs. Therefore, we view the proposed approach as orthogonal to goal-conditional RL and they could potentially be integrated together to bring further improvements.

**Model-based RL and Plan-to-Explore.** Model-based RL is another large family of methods that has been explored from many perspectives (*e.g.* Sutton, 1991; Silver et al., 2016; Srinivas et al., 2018; Chua et al., 2018; Agarwal et al., 2018; Lowrey et al., 2019; Hoppe et al., 2019). *i) Learning with latent dynamics.* While some works may require a given model (Silver et al., 2016; Lowrey et al., 2019), many recent methods work with a latent dynamics model (*e.g.* Oh et al., 2017; Silver et al., 2017; Schrittwieser et al., 2019; Hafner et al., 2020). Our work is related to Oh et al. (2017); Silver et al. (2017); Schrittwieser et al. (2019) in the sense that our plan value function is trained by predicting future values, and related to Hafner et al. (2020) in the way of policy training by maximizing values. However, these approaches do not leverage their implicit models for temporally extended exploration, which is the focus of our work, thus are orthogonal to our work. Also, our value function has a dedicated latent dynamics model that is not shared with the plan generator, which offers more flexibilities in structural design (*e.g.* the usage of a structural prior Eqn.(4) in plan generator) and learning (each is responsible for its own objective). Therefore, the overall algorithmic structure resemble standard model-free RL methods, making the transition between them more natural. *ii) Integrating model into model-free RL.* There are many possible ways to integrate model into model-free RL methods (*e.g.* Sutton, 1991; Levine & Koltun, 2013a;b; Racanière et al., 2017;

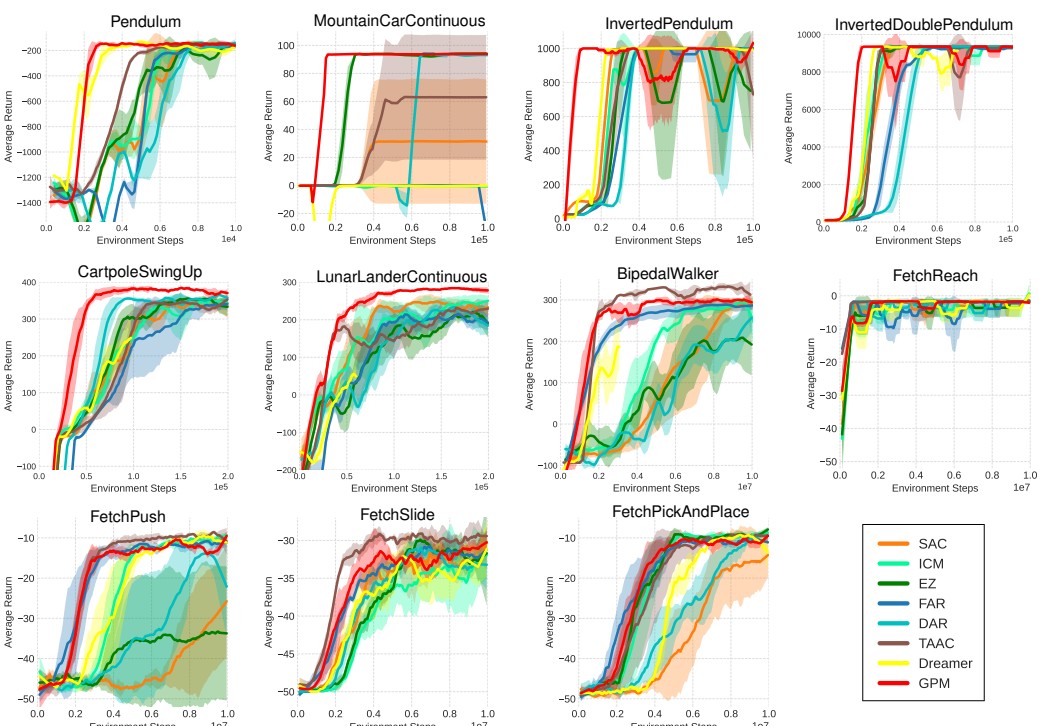

Figure 3: **Return Curves** on benchmark continuous control tasks for different methods.

Srinivas et al., 2018). Imagination-augmented agent (Racanière et al., 2017) uses the predictions from a model as additional contexts for a model-free policy. Srinivas et al. (2018) embeds differentiable planning within a goal-directed policy and optimizes the action plan through gradient descent. Our work is different in that it uses a plan generator to directly generate action sequences conditioned on the current state, much like a model-free policy, without using additional context from a model-based prediction (Racanière et al., 2017) or requiring expensive gradient based plan generation (Srinivas et al., 2018). ***iii) Plan-to-explore.*** The general idea of plan-to-explore is broadly applicable and has appeared in many categories including hierarchical (*e.g.* Konidaris et al., 2010; Pertsch et al., 2020), goal-conditional (*e.g.* Ren et al., 2019; Pong et al., 2020; Pitis et al., 2020) as well as model-based RL (*e.g.* Pathak et al., 2017; Henaff, 2019; Shyam et al., 2019; Sekar et al., 2020). In this work, we focus on improving standard model-free RL methods by generating coordinated action sequence for temporally extended exploration, which has a different focus and complementary to these approaches.

## 5 EXPERIMENTS

**Baselines Details.** We implement and compare the performance of the proposed method with seven baseline methods (Xu et al.).: *i)* `SAC`: the Soft Actor Critic (Haarnoja et al., 2018) algorithm which is a representative model-free RL algorithm; *ii)* `EZ`: temporally-extended $\epsilon$-Greedy exploration ($\epsilon z$-Greedy) (Dabney et al., 2021), which repeat the sampled action for a random duration during exploration; *iii)* `FAR`: Fixed Action Repeat (Mnih et al., 2015), which repeats the policy action for a fixed number of steps; *iv)* `DAR`: Dynamic Action Repeat with TempoRL (Biedenkapp et al., 2021), which improves over `FAR` with a action-conditional policy for predicting a duration to repeat, outperforming non-conditional alternatives (Sharma et al., 2017); *v)* `TAAC`: Temporally Abstract Actor Critic (Yu et al., 2021), which uses a switching policy for determining whether to repeat the previous action, instead of generating a duration to repeat beforehand as in `DAR` (Biedenkapp et al., 2021); *vi)* `ICM`: Intrinsic Curiosity Module (Pathak et al., 2017), which is a model-based method for reward bonus based exploration. *vii)* `Dreamer`: a model-based RL method which trains the policy by back-propagation through the learned latent dynamics (Hafner et al., 2020). `SAC` is used as the backbone algorithm for *ii)~vi)*. More results and resources are available on the project page. [1]

---

[1] https://sites.google.com/site/hczhang1/projects/generative-planning

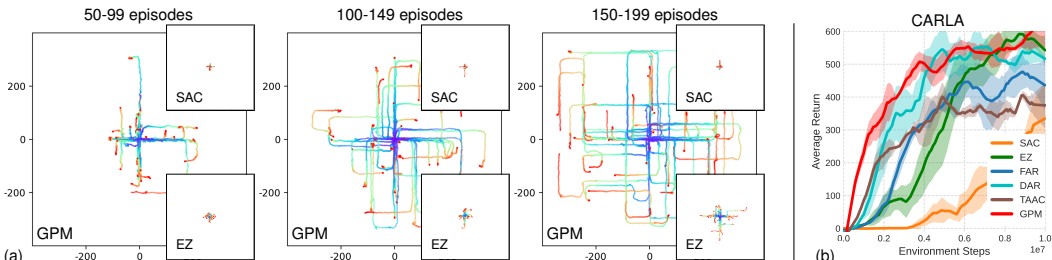

Figure 4: **Exploration Trajectory Visualization and Return Curves**. (a) Each figure shows collectively the 50 rollout trajectories during training for the range depicted above. Trajectories are plotted relative its start position for easier visualization (unit:meter). The chronological order of positions in the trajectory is encoded in the same way as Figure 5. Visually, an effective agent should push the position of the red points as far as possible along its route. All plots including thumbnails are rendered at the same scale relative to their axes. (b) Return curves of different methods.

## 5.1 EXPERIMENTS ON CONTINUOUS CONTROL

**Benchmark Results.** We conduct experiments on 12 benchmark tasks and the results are shown in Figure 3 (image-based CARLA results are deferred to Figure 4). It is observed that vanilla **SAC** can achieve reasonable performance on most of the tasks, albeit with a large performance variance and high sample complexity. **FAR** shows better performance than **SAC** on some tasks (*e.g.* BipedalWalker and FetchPush), but no clear improvements (*e.g.* InvertedPendulum) or even worse performance on some other tasks (*e.g.* CartpoleSwingUp). This is potentially because of the relatively more restrictive policy with a fixed repetition scheme. **EZ** outperforms **SAC** on some tasks (*e.g.* MountainCarContinuous and FetchPickAndPlace) and retain mostly comparable performance to **SAC** on others, demonstrating the contribution of incorporating temporal consistency to the exploration policy. **ICM** and **Dreamer** improves over **SAC** on some tasks but not always. Both **DAR** and **TAAC** show consistent improvements over **SAC** due to their flexibility on leveraging action repeat to the extend where it is useful. The proposed **GPM** demonstrates further improvements in terms of sample efficiency on some of the tasks, with final performance matching or outperforming those of the baseline methods. Apart from state-based control, **GPM** also delivers strong performance in a more challenging image-based control task (*c.f.* Figure 4), as detailed in Section 5.2.

**State Trajectory Visualization.** To understand the behavior of different algorithms, we further visualize the state visitation distribution and its evolution during training (Figure 12). It shows that during the initial phase of training, the state coverage of the methods with extended temporal exploration is much higher than that of **SAC**, which are consistent with the results reported in (Dabney et al., 2021). Furthermore, as training progresses, **GPM** is able to find the high rewarding state more quickly, visually depicting its improved exploration performance and sample efficiency.

## 5.2 END-TO-END AUTONOMOUS DRIVING WITH EMERGED INTERPRETABLE PLANS

Autonomous Driving (AD) is an important application for learning-based algorithms. Although the end-to-end approaches have showed encouraging results (*e.g.* Bojarski et al., 2016; Toromanoff et al., 2020), one great limitation is their limited interpretability. Although **GPM** is also an end-to-end approach, it has improved interpretability, in addition to its better performance. It is important to note that this ability of generating interpretable plans is emerged from pure reinforcement learning, without direct supervisions. This is in clear contrast to the methods from the decomposed paradigm, which requires additional labels as direct supervision for training (Phan-Minh et al., 2020; Rhinehart et al., 2020). We use CARLA (Dosovitskiy et al., 2017) to setup a challenging image-based driving task. We set the plan length as $L = 20$ to enable more perceivable results when visualized. More details including observations, actions, rewards and network structure are presented in Appendix A.5.

**Behavior and Performance Comparison**. The exploration behavior comparison of **GPM** with **SAC** and **EZ** are shown in Figure 4 (a). It can be observed that the effective range of exploration of a **SAC** agent increases slowly and is quite limited, demonstrating its inefficiency in exploration. **EZ** shows enhanced exploration ability over **SAC**, as depicted by the enlarged range of explored areas. **GPM** shows a further improved exploration ability. More specifically, **GPM** shows some directional movement ability during the initial phase of training. It further shows reasonable turning behavior after about 100 training episodes and more complex behavior with multiple turns after about 150 training episodes. In terms of performance, it can be observerd from Figure 4 (b) that **GPM** learns faster than other baseline methods, with final performance better than or competitive to other methods.

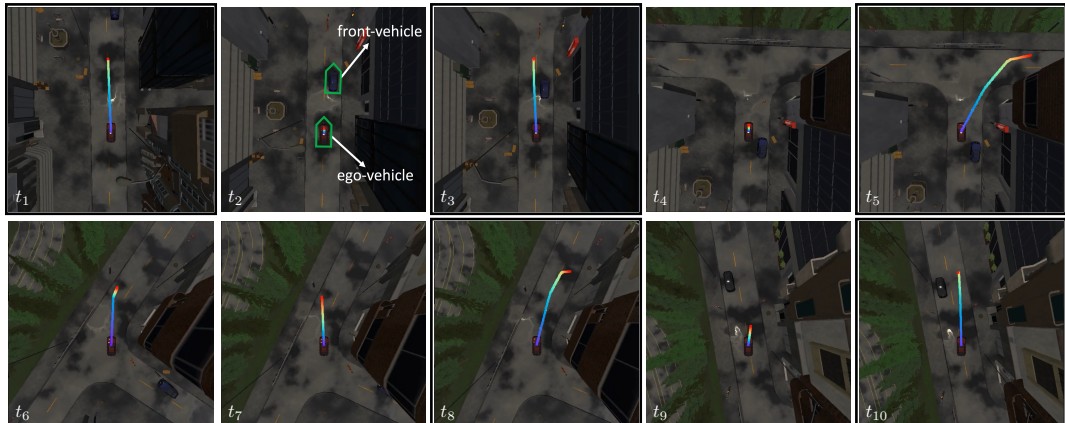

Figure 5: **Evolution of GPM Plans**: (a) initially the plans are nearly uniformly distributed across all the directions; (b) forward-moving plans appear to take a larger portion afterwards; (c) further training shows the emergence of plans with more adjustments on speed, indicated by shorter length plans; (d) later, more curved turning plans and plans with increased length appear; (e) at the end of training, the agent can drive smoothly, therefore forward moving and turning plans take the majority.

Figure 6: **Plan Visualization within an Episode**. Note that the agent is trained with a frontal camera capturing $64{\times}64$ RGB images. The top-down view at a higher resolution is only used for visualization. The plan is color-coded as in Figure 5. Frames switching to new plans are enclosed with a black box. At $t_1$, a roughly straight plan is generated, and it is committed to for multiple time steps till $t_2$. At the next time step $t_3$, the agent switches to a new plan, passing a car in the front. This plan is committed to by the agent ($t_3 \rightarrow t_4$) and at $t_5$, another new plan curved towards the turning direction is adopted. $t_6$ and $t_7$ show two frames while committing to the $t_5$-plan for a right turn. After that, the old plan is replaced by a new plan at $t_8$, executed till $t_9$ and is replaced by another one at $t_{10}$.

**Emerged and Evolved Planning**. The evolution of the plans from **GPM** is visually shown in Figure 5. It shows the emergence of meaningful plans and the evolution of them along the training process. This naturally depicts several advantages of **GPM** over standard end-to-end AD systems: *i)* it has more interpretable plans, which can serve as a media for communicating the intent of the algorithm; *ii)* the evolution of the algorithm behavior during training is more perceivable; *iii)* the plans, with grounded meaning of spatial-temporal trajectories, offer the opportunity of further adjustment according downstream requirements in the AD pipeline, which could potentially narrow down the gap between end-to-end and decomposed approaches and provide an opportunity to integrate. The plans from **GPM** together with the visual context images are provided in Figure 6. It can be observed that **GPM** plans are interpretable and meaningful. Note that this a type of *emerged planning* ability as there is no direct supervision on it but purely acquired through interactions with the environment.

## 6 CONCLUSION AND FUTURE WORK

We explore a perspective on using planning for inducing temporally extended, flexible and intentional action sequence for exploration in a generative fashion. We present a Generative Planning method as an attempt towards this goal. The proposed approach features an inherent mechanism for generating temporally coordinated plans for exploration, while retaining the major features and advantages of typical model-free RL algorithms. We hope this work can serve as an initial encouraging step that can inspire more developments in this direction. For future work, the quality of the plans can potentially be further improved by learning a more refined cost with the help of additional data (Ziebart et al., 2009; Wulfmeier et al., 2016; Choi et al., 2019; Rosbach et al., 2019). While we have mainly focused on continuous control tasks in this work, it is possible to generalize GPM to handle discrete actions (Tennenholtz & Mannor, 2019). The network architecture also provides another axis for potential improvements. We leave the exploration of these interesting directions as our future work.

**Acknowledgements.** We would like to thank Jingchu Liu, Lisen Mu, Jerry Bai, Le Zhao, Break Yang, Lingfeng Sun and anonymous reviewers for discussions and feedback on the project and paper. We would also like to thank the Horizon AI platform team for infrastructure support.

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

## A APPENDIX

### A.1 GPM ALGORITHM

The overall procedure of the proposed method is presented in Algorithm 1.

---

**Algorithm 1** Generative Planning Method (GPM)

---

**Initialize:** $\theta, \phi$
$\phi' \leftarrow \phi, \mathcal{D} \leftarrow \emptyset$
**for** each iteration **do**
  **for** each environment step **do**
    $\tau_{\text{old}} = \emptyset$ if $|\tau_{\text{old}}|$ is 0 else $\tau_{\text{old}} \leftarrow \rho(\tau_{t-1})$     $\triangleright$ time-forward the old plan
    $\tau_{\text{new}} \sim \Pi(s_t)$     $\triangleright$ generate a new plan
    $m = 1$ if $l_{\text{old}} = |\tau_{\text{old}}|$ is 0 else $m \sim \text{Categorical}([\epsilon, \mathbb{Q}_{l_{\text{old}}}(s, \tau_{\text{new}}) - \mathbb{Q}_{l_{\text{old}}}(s, \tau_{\text{old}})])$
    $\tau_t = (1 - m) \cdot \tau_{\text{old}} + m \cdot \tau_{\text{new}}$     $\triangleright$ select the actual plan from old and new plans
    $a_t \leftarrow \omega(\tau_t)$     $\triangleright$ extract the primitive action for the current step
    $s_{t+1} \sim p(s_{t+1}|s_t, a_t)$
    $\mathcal{D} \leftarrow \mathcal{D} \cup \{(s_t, a_t, r(s_t, a_t), s_{t+1})\}$     $\triangleright$ save transition to replay buffer
  **end for**
  **for** each train step **do**
    $\phi \leftarrow \phi - \lambda \nabla_\phi J_{\mathbb{Q}}(\phi)$     $\triangleright$ plan value update
    $\theta \leftarrow \theta - \lambda \nabla_\theta J_\Pi(\theta)$     $\triangleright$ plan generator update
    $\alpha \leftarrow \alpha - \lambda \nabla_\alpha J(\alpha)$     $\triangleright$ temperature update
    $\epsilon \leftarrow \epsilon - \lambda \nabla_\epsilon J(\epsilon)$     $\triangleright$ replan threshold update
    $\phi' \leftarrow \eta\phi + (1 - \eta)\phi'$     $\triangleright$ target parameter soft-copy
  **end for**
**end for**

---

$J(\alpha) = \mathbb{E}_{a_t = \omega(\tau), \tau \sim \Pi} \left[ -\alpha \log \pi_0(a_t|s_t, \alpha) - \alpha\bar{\mathcal{H}} \right]$, where $\pi_0$ denotes the policy corresponding to the first step of the plan, which is similar to the one used in Haarnoja et al. (2018). $\bar{\mathcal{H}}$ is the target entropy set in the same way as SAC (Haarnoja et al., 2018) for controlling the degree of stochasticity of the policy by adjusting the temperature $\alpha$.

### A.2 DETAILS ON $\rho$ AND $\omega$ OPERATORS

Here we summarize in Table 1 the meaning and implementations of some entities as well as operators that are used in this work to further help understanding.

| Entity/Operator | Meaning/Implementation | |
|---|---|---|
| | Gym Task | CARLA Task |
| $\tau = [a_0, \cdots, a_i, \cdots]$ | $a_i$ denotes a primitive action | $a_i$ denotes a setpoint in ego-coordinate |
| $\rho(\tau)$ | $\rho(\tau) = \mathcal{T}^{\text{shift}}(\tau) = \tau[1:]$ | $\rho(\tau) = \mathcal{T}^{\text{world2ego}} \circ \mathcal{T}^{\text{shift}} \circ \mathcal{T}^{\text{ego2world}}(\tau)$ |
| $\omega(\tau)$ | $\omega(\tau) = \tau[0]$ | |

Table 1: **Details on $\tau$, $\rho$ and $\omega$.**

In Table 1, $\mathcal{T}^{\text{ego2world}}$ is an operator for transforming the plan from ego to world coordinate and $\mathcal{T}^{\text{world2ego}}$ the reverse operator.

### A.3 NETWORK STRUCTURE AND DETAILS

Network details for different tasks are summarized in Table 2, where we use $[K] \times L$ to denote an MLP with $L$ FC layers and each with the output size of $K$. For baseline methods, they use the network structure as shown in Table 2 for both actor and critic networks. For **GPM**, we implement the plan generator and plan value network as in Figure 2. The plan generator first use an MLP that has the same structure as baseline methods as shown in Table 2 for encoding the observation $s$ to a feature vector $z$. $z$ is used as the input of the Normal Layer, which uses FCs for generating mean and std vectors of a diagonal Gaussian distribution (Haarnoja et al., 2018). $z$ is passed through a FC layer to the initial state of a GRU network (plan RNN), with its hidden size the same as the single

layer size in Table 2. Action is detached before inputting to the GRU to to enforce the training of GRU and improve training stability. A shared FC layer is used for generating the actions after the first step, taking the output from plan RNN as input. The plan value network is implemented with LSTM (value RNN) with hidden size the same as the single layer size in Table 2. An MLP decoder with the same structure as those used in other baseline methods (Table 2) are used for plan value estimation, taking the output from value RNN as decoder input. The first step uses one such decoder with parameters not shared with others. The rest of the steps uses a shared one. The reason for this is that the conditioning state for the first step is different from the rest. ReLU activation is used for hidden layers. Following standard practice, the minimum over the outputs of two value networks is used to mitigate value overestimation (Fujimoto et al., 2018; Haarnoja et al., 2018).

| Task | Network (actor/critic) | learning rate | batch size | plan length | # parallel actors |
|---|---|---|---|---|---|
| Pendulum | $[100] \times 2$ | $5e^{-4}$ | 64 | 3 | 1 |
| InvertedPendulum | $[256] \times 2$ | $1e^{-4}$ | 256 | 3 | 1 |
| InvertedDoublePendulum | $[256] \times 2$ | $1e^{-4}$ | 256 | 3 | 1 |
| CartPoleSwingUp | $[256] \times 2$ | $1e^{-4}$ | 256 | 3 | 1 |
| LunarLanderContinuous | $[256] \times 2$ | $1e^{-4}$ | 256 | 3 | 1 |
| MountainCarContinuous | $[256] \times 2$ | $1e^{-4}$ | 256 | 10 | 1 |
| BipedalWalker | $[256] \times 2$ | $5e^{-4}$ | 4096 | 3 | 32 |
| Fetch | $[256] \times 3$ | $1e^{-3}$ | 5000 | 3 | 38 |
| CARLA | observation_encoder $\to [256]$ | $1e^{-4}$ | 128 | 20 | 4 |

Table 2: **Details on network structure and hyper-parameters**. More details on the observation encoder for CARLA is provided in Appendix A.5.2 and Figure 7.

## A.4 GYM TASK DETAILS

The Gym tasks used in this paper and their corresponding Gym environment names are summarized in Table 3. **Fetch** tasks are from Plappert et al. (2018a).

| Task | Gym Environment Name |
|---|---|
| Pendulum | Pendulum-v0 |
| InvertedPendulum | InvertedPendulum-v2 |
| InvertedDoublePendulum | InvertedDoublePendulum-v2 |
| MountainCarContinuous | MountainCarContinuous-v0 |
| CartPoleSwingUp | CartPoleSwingUp-v0 [2] |
| LunarLanderContinuous | LunarLanderContinuous-v2 |
| BipedalWalker | BipedalWalker-v2 |
| FetchReach | FetchReach-v1 |
| FetchPush | FetchPush-v1 |
| FetchSlide | FetchSlide-v1 |
| FetchPickAndPlace | FetchPickAndPlace-v1 |

Table 3: **Tasks and gym environment names**.

## A.5 CARLA TASK DETAILS

The CARLA driving task is implemented based on the open source Car Learning to Act (CARLA) simulator (Dosovitskiy et al., 2017). More details on sensor/observation/action/reward, network structure *etc.* are presented in the sequel.

### A.5.1 SENSORS AND OBSERVATIONS

The following types of sensors are used in CARLA driving task (*c.f.* Figure 7):

- **Camera**: frontal camera capturing RGB image of size 64×64×3;
- **IMU sensor**: 7D vector containing acceleration (3D), gyroscope (3D), compass (1D), providing some measurements on the status of the current vehicle;

---

[1]https://github.com/angelolovatto/gym-cartpole-swingup

- **Velocity**: 3D vector containing velocity relative to ego-coordinate in m/s;
- **Navigation**: the collection of the positions of 8 waypoints $d$ meters away from the ego vehicle along the route, with $d \in \{1, 3, 10, 30, 100, 300, 1000, 3000\}$ meters. The positions of the waypoints are represented in 3D ego-coordinate. The sensory reading is a 24D vector;
- **Goal**: 3D position in ego-coordinate representing the target position the ego car should navigate to and stop at;
- **Previous action**: 2D vector representing the previous action. More details on the action space are provided in Appendix A.5.3.

The delta time between two time steps is $\Delta t = 0.05$s, *i.e.*, the perception and control rate is 20 Hz.

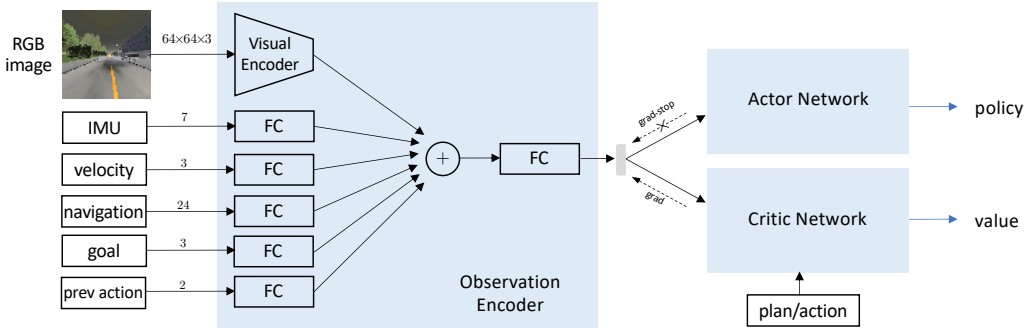

Figure 7: **Network structures** for CARLA task.

### A.5.2 CARLA NETWORK STRUCTURE

The network structure used in CARLA task is shown in Figure 7. This is the common structured shared by all the methods including **GPM**. Given the observation containing multi-modality signals, an observation encoder is first used to fuse them into a low-dimensional feature vector. This is achieved by processing each modality of signals with a dedicated encoder with the same dimensionality for the outputs. More specifically, the visual encoder is a ResNet-18 based CNN. Each of other modalities are encoded by a fully connected layer. The outputs from each of the encoder are fused together with a sum operator followed by a FC layer to produce the feature vector as input to downstream networks. The observation encoder is shared by both the actor and critic network. During training, the the gradient from the actor is stopped and the observation encoder is only trained by value learning.

### A.5.3 CARLA CONTROLLER AND ACTION SPACE

The CARLA environment with its controller interface is illustrated in Figure 8. It takes as input a setpoint in ego-coordinate, specifying an immediate position to go to for the current step. For the driving case, the setpoint can be represented in the 2D spatial ego-coordinate. Therefore there are two dimensions ($x$: lateral, $y$: longitudinal) in the action space, with $a = (x, y) \in [-20, 20] \times [-20, 20]$. The actor controls the movement of the car by generating 2D action as the setpoint. Target direction and speed will be computed based on the setpoint and ego-position (origin), which will be passed to a PID-controller to produce control signals including throttle, steer, brake and reverse. These signals will be used by the simulator to generate the next state of the car. As part of the environment, a shrinkage is applied to the setpoint before computing the control signals:

$$\tilde{a} = a \cdot \frac{\max(\|a\| - \delta, 0)}{\|a\|}. \tag{8}$$

This is used to balance the portion of the braking behavior *w.r.t.* others in the control space. $\delta = 3$ is used in this work.

### A.5.4 REWARD

The reward in CARLA environment is consisted of the following components:

Figure 8: **CARLA environment and action space**.

- **Route distance reward**: this reward is computed as the distance traveled along route from the previous to the current time step.

- **Collision reward**: this is computed as $\min(\text{collision\_penalty}, 0.5 * \max(0, \text{episode\_reward}))$, where $\text{collision\_penalty}=20$, $\text{episode\_reward}$ denotes the accumulated rewards up to now within the current episode. Issued only once for a contiguous collision event.

- **Red-light reward**: this is computed as $\min(\text{red\_light\_penalty}, 0.3 * \max(0., \text{episode\_reward}))$, where $\text{red\_light\_penalty}=20$. Issued only once for a contiguous red light violation event.

- **Success reward**: a success reward of 100 will be received by the agent (ego-car) if its position is within 5 meters to the goal position and its speed is less than $10^{-3}$ m/s.

### A.5.5 EPISODE TERMINATION CONDITIONS

The episode terminates if either of the following conditions are satisfied:

- success: the car's position is within 5 meters to the goal position and its speed is less than $10^{-3}$ m/s.

- the maximum number of allowed time steps $K$ is reached: this number is computed as $K = \frac{\text{route\_length}}{\text{min\_velocity} \times \Delta t}$, where the $\text{min\_velocity} = 5$ m/s and $\Delta t = 0.05$s. The state is marked as a non-terminal state, allowing bootstrapping for value learning at this state;

- stuck at collision: if the car is involved in the same collision event and does not move more than 1 meter after 100 time steps. The state is marked as a non-terminal state, allowing bootstrapping for value learning at this state.

### A.5.6 MAP AND EXAMPLE ROUTES

For CARLA driving task, we use Town01, which is a town with representative layout consisting a number of T junctions and intersections. The layout of Town01 is shown in Figure 9 (a). The size of the town is about $410\text{m} \times 344\text{m}$. We also spawn 20 other vehicles controlled by autopilot and 20 pedestrians to simulate typical driving conditions involving not only static objects but also dynamic ones.

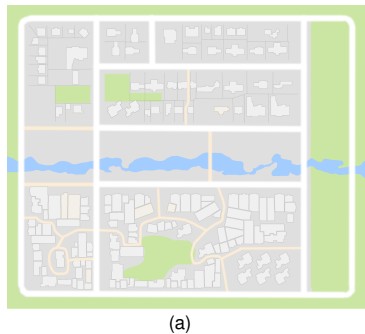
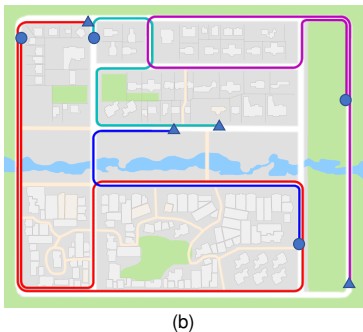

(a)                        (b)

Figure 9: **Map of and Example Routes**. (a) Map of Town01 (Dosovitskiy et al., 2017). (b) Map with some example routes overlaid. Different routes are rendered with different colors. The beginning of each route is denoted with a $\triangle$ symbol and the end of the route is denoted with a $\bigcirc$ symbol.

The goal location is randomly selected from the set of waypoints. The agent is spawned to a random valid waypoint location at the beginning of each episode. A waypoint is valid if it is on the road and is at least 10 meters away from any other actor in the world (vehicles and pedestrians). Some examples routes are shown in Figure 9(b). As can be observed from the figure, the routes constructed in this way have a reasonable amount of diversity and cover representative driving scenarios. For example, different routes could include different number of turns, directions for turning, static objects (as the routes are different) and also different dynamic objects (*e.g.* other vehicles) at different space-time locations. The task of the agent is to navigate to the goal position following the route and stay still after arrived at the goal location.

## A.6 MORE IMPLEMENTATION DETAILS

### A.6.1 PLAN SAVING, SAMPLING AND GENERATION

In practice, for plan value learning, for each training iteration, we sample a batch of (state, plan, rewards, next-plan-state) tuple $\{s_t^i, \tau_t^i, \mathbf{r}_t^i, s_n^i\}_{i=1}^N$ from the replay buffer. More concretely, for each state $s_t^i$ randomly sampled from the replay buffer, we sample an associated plan $\tau_t^i$ as the actions starting from the current step of $s_t^i$ to a variable number of future steps uniformly sampled in the range of 1 and maximum plan length $L$, *i.e.* $\tau_t^i = [a_t^i, \cdots, a_{t+l-1}^i]$ where $l \sim \text{uniform}(1, L)$. The reward vector is consisted of the rewards along the plan $\mathbf{r}_t^i = [r_{t+1}^i, \cdots, r_{t+l}^i]$. The plan terminal state $s_n^i$ is the state at the time step after the last step of $\tau_t^i$, *i.e.* $s_n^i = s_{t+l}^i$. Therefore, in effect, we use sampled plan for plan value learning. By constructing plans in this way, we have maximal flexibility of leveraging the action sequences from replay buffer with minimal assumption on the plan distribution. During training, $l_{\text{commit\_target}}$ is set to $0.5L$.

Note that in the case of driving (and other similar scenarios), since the plan is a spatial trajectory in the ego-coordinate, and the plan update operator in this case is $\rho(\tau) = \mathcal{T}^{\text{world2ego}} \circ \mathcal{T}^{\text{shift}} \circ \mathcal{T}^{\text{ego2world}}(\tau) = \mathcal{T}^{\text{world2ego}} \circ \mathcal{T}^{\text{shift}}(\tau')$ (*c.f.* Table 1). To enable the construction of plans from action sequences sampled from the replay buffer (not necessarily aligned with the plan boundary during rollout), we need to save the $a' = \omega(\tau')$ into replay buffer, *i.e.* $\mathcal{D} \leftarrow \mathcal{D} \cup \{(s_t, a_t', r(s_t, a_t), s_{t+1})\}$. $a' = \omega(\tau')$ actually denotes the setpoint in world coordinate. In this case, since the actions saved in replay buffer are in world coordinate, we can construct plans in world coordinate by by sampling sequences from the replay buffer, in the same way as described above, *i.e.* $\tau_t' = [a_t, \cdots, a_{t+l-1}']$ where $l \sim \text{uniform}(1, L)$. Then the plans in ego-coordinate can be obtained as $\mathcal{T}^{\text{world2ego}}(\tau_t')$. For the standard Gym tasks, since both $\mathcal{T}^{\text{world2ego}}$ and $\mathcal{T}^{\text{world2ego}}$ are identity operators, we have $\tau' \equiv \tau$, meaning we can simply save the current primitive action $a = \omega(\tau)$ into replay buffer.

A scaling factor can be used for the replanning distribution $\text{Categorical}(\frac{z}{\kappa})$. We simply use $\kappa=1$. For testing the plan generator after training, the mode of the policy is used, by taking the mean of the Gaussian policy used in the plan generator and taking the mode of $\text{Categorical}(z)$ for replanning.

### A.6.2 PLAN GENERATOR IN AUTONOMOUS DRIVING

We implement $f$ used in plan generator as follows

$$a_{t+i} = a_{t+i-1} + \Delta_{a_{t+i-1}} + g(h_{t+i}, a_{t+i-1}, \Delta_{a_{t+i-1}}) \qquad i \in [1, n-1] \tag{9}$$

where $\Delta_{a_{t+i-1}} = (a_{t+i-1} - a_{t+i-2})$. This actually incorporates a linear prior into $f$, which is useful for tasks with position-based controls. A smoothing cost is also applied to the plan to minimize the maximum absolute value of its third order derivative $\max |\dddot{\tau}|$ (Kyriakopoulos & Saridis, 1988).

In the most basic form, the recurrent plan generator can run at the scale of time-step in order to generate the full plan. In practice, it can be adjusted according to the actual situation, and running at a coarser scale, which has the advantage of lower-computational cost. For example, we can let the policy only output a few number ($k$) of *anchor points* and interpolate between them to get the full plan using a particular form of interpolation methods. Linear interpolation is used in this work.

## A.7 MORE RESULTS

### A.7.1 ABLATION RESULTS ON PLAN LENGTH AND TARGET COMMITMENT LENGTH

**Impact of Plan Length.** The results with different plan lengths are shown in Figure 10(a). It is observed that the overall the performance is relatively insensitive to plan length, although the performance could be further optimized by tuning this hyper-parameter.

**Target Commitment Length.** The results with different target commitment lengths are shown in Figure 10(b). The results indicate that the performance of the proposed method is relatively stable across different target commitment lengths, potentially due to its ability in generating plans with adaptive shapes.

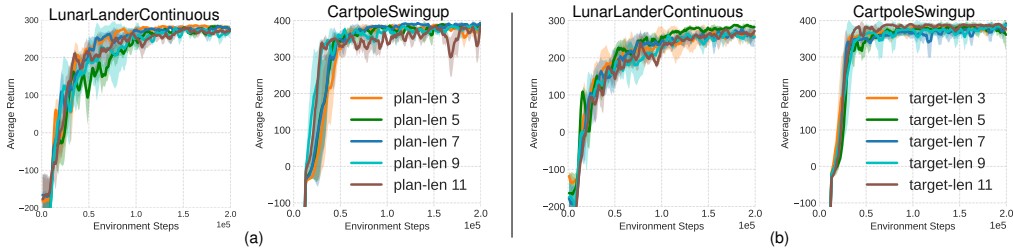

Figure 10: **Ablations** on (a) plan length and (b) target commitment length. For (a) the target commitment length is set as 1.5. For (b) the plan length is set as 5.

### A.7.2 OBSERVATIONAL DROPOUT: THE IMPORTANCE OF ADAPTIVE PLANS

Here we compare the proposed method and action repeat under the setting where a fixed replanning duration is used. For action repeat, this corresponds to the **FAR** algorithm. For **GPM**, we also execute the full plan before switching (denoted as **GPM-commit**). This actually an instance of *observational dropout* (Freeman et al., 2019), which requires the algorithm to extrapolate controls generated from available observations to those time steps where observations are unavailable. The results are shown in Figure 11. This results highlight the importance of *adaptive* form of plans on performance. Fixed form of plan as used in **FAR** might be applicable to certain types of tasks (*e.g.* CartPoleSwingUp). However, for other tasks, it might be less appropriate and leads to significant performance decreases with increasing plan length (*e.g.*, LunarLanderContinuous). In contrast, the proposed approach shows stable performance across a wide range of plan lengths. By contrasting the performance of **FAR** with **GPM-commit**, we highlight the importance on the adaptiveness of the plans.

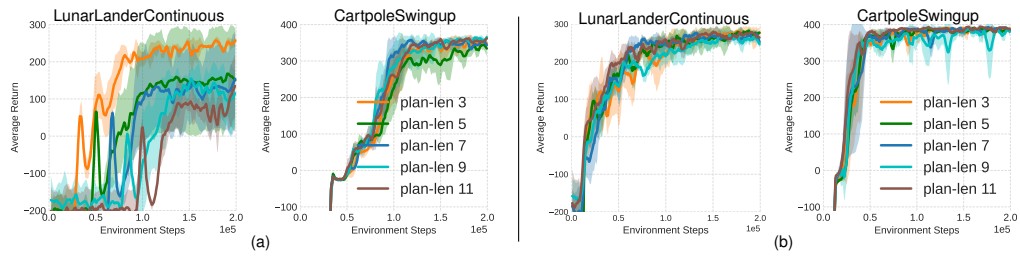

Figure 11: **Observational Dropout** for (a) **FAR** (b) **GPM-commit**. Here both approaches generate plans of fixed length and execute them to the end before generating the next plan.

### A.7.3 MPC V.S. GPM

MPC has been typically used together with planning for obtaining the control policy (Lowrey et al., 2019). This strategy is appropriate when a model is given (Lowrey et al., 2019). However, for exploration in RL, MPC is less desirable as it essentially switches the plan at every step, not leveraging the temporal property of the plans, as shown on the right, by comparing with a variant of using MPC together with GPM (GPM-MPC).

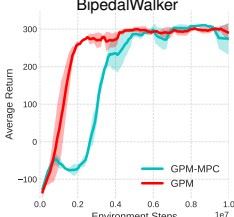

### A.7.4 EXPLORATION TRAJECTORY VISUALIZATION AND COMPARISON

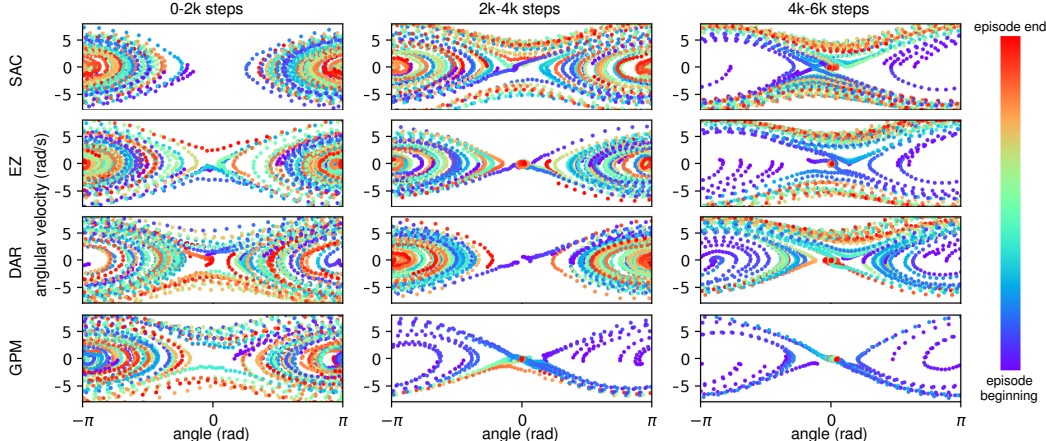

Figure 12: **Visualization of State Visitation and Evolution** during training on `Pendulum`. Each figure shows collectively all the rollout trajectories within the range of steps depicted on the top. Since the most rewarding state is $(0, 0)$, an effective policy should quickly move close to this state.

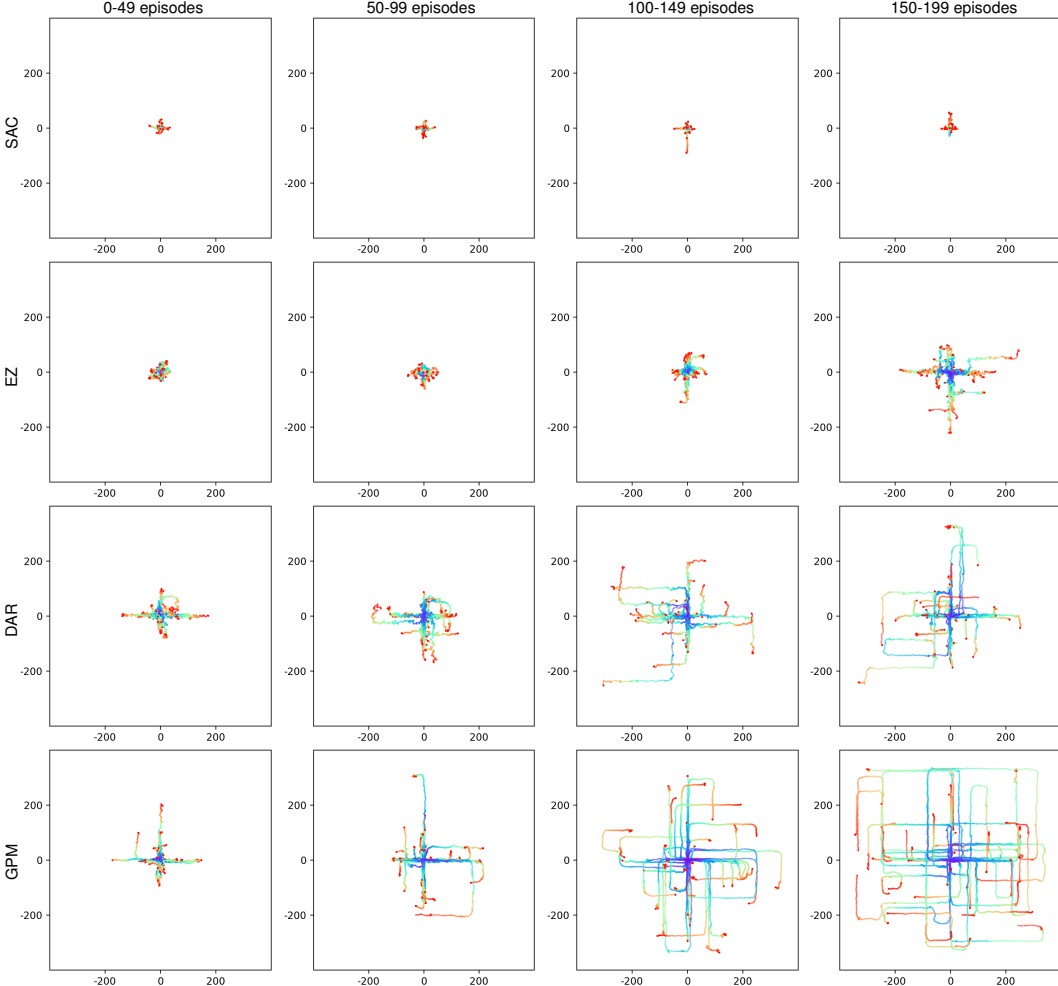

Figure 13: **Exploration Trajectory Visualization**. Trajectory are plotted relative its start position.

