# OpenReview forum: "Generative Planning for Temporally Coordinated Exploration in Reinforcement Learning"
_ICLR.cc/2022/Conference — ICLR 2022 Spotlight_

### Official Review · Reviewer_356g · 2021-10-30

**Correctness:** 4
**Technical Novelty And Significance:** 4
**Empirical Novelty And Significance:** 3
**Recommendation:** 8
**Confidence:** 4

**Main Review:**

This paper is well-written and easy to follow. The motivation is clear and important: how to improve exploration for model-based RL. The proposed method of planning in a short term and encouraging using the old plan in order to encourage temporally extended exploration is neat and interesting. The evaluations are solid, comparing with 5 baselines on 8 benchmarks. Overall, this is a good paper.

Strength:
- Well-written and easy to follow
- The motivation is clear: how to improve exploration for model-based RL
- Proposed technique is interesting: learning a recurrent model to generate short-term plans at each time step, and only switching to the new plan if it is a lot better than the old plan.
- Solid evaluations. GPM is compared with 5 baselines on 8 benchmarks, and experiments show that GPM converges faster than the other baselines. Further analyses show that GPM indeed explores more effectively, and it also generates interpretable short-term plans

Questions:
- What is the computational overhead of GPM compared with the baselines? Since it needs to generated short-term plans at each time step, does it take longer on average to run each training step?
- It makes sense to encourage reusing the old plan during training since it encourages exploration. When the learned policy is deployed in test time, is it better to remove this bias towards the old plan? I would like to see a discussion on this from the authors.


**Summary Of The Paper:**

This paper proposes a method called Generative Planning (GPM), which aims to improve exploration for model-free RL. GPM learns a recurrent model to generate short term plans at each time step, and only decides to switch to the new plan if it is a lot better than the old plan. This encourages temporally extended explorations, and also does it in an adaptive manner. Experiments on a set of continuous control benchmarks show that GPM is able to converge faster than prior approaches, explore more effectively, and generate interpretable short-term plans.


**Summary Of The Review:**

This paper is well-written and well-motivated. The proposed technique is interesting and the experimental section is strong. Overall, this is a good paper and I recommend acceptance.

---

> ### Author Response · Authors · 2021-11-23
> **Response to Reviewer 356g**
>
> We thank the reviewer for the positive review and insightful comments.
>
> Responses to the questions are below:
>
> **Q1: "What is the computational overhead of GPM compared with the baselines ... does it take longer on average to run each training step?"**
>
> Your understanding is correct. GPM is about 2~3 times slower than SAC on average for each train step (including unroll). This is because of the additional computations including plan generation and plan value computation during rollout, as well as the sequence based training such as plan value training and plan generator training, which involve recurrent computation instead of feedforward computation.
>
> It's possible to improve speed while maintaining performance by further improving network architectures, e.g. using  causal convolution. This is orthogonal to the focus of the current paper and we will explore it in the future.
>
> **Q2: "It makes sense to encourage reusing the old plan during training since it encourages exploration. When the learned policy is deployed in test time, is it better to remove this bias towards the old plan? I would like to see a discussion on this from the authors."**
>
> In most of the continuous control cases, since states in a reasonable span of future are predictable from the current state, the plan generator can acquire some predictive abilities for future actions. In this case, even if we do not replan for each time step, the performance won't be compromised (Appendix A.7.2 observational dropout).
>
> On the other hand, for our current design, simply use the new plan at each time could have some computational benefits at test time, since computation for the plan values can be avoided.

---

> > ### Comment · Reviewer_356g · 2021-11-26
> > **Re: response**
> >
> > Thank you for the response. This addresses my questions. I have no further questions at this point.

---

### Official Review · Reviewer_3VLM · 2021-10-31

**Correctness:** 4
**Technical Novelty And Significance:** 4
**Empirical Novelty And Significance:** 4
**Recommendation:** 8
**Confidence:** 4

**Main Review:**

Strengths:
- the method is (to my knowledge) a novel approach that I see as sitting somewhere between model-based and model-free.
  It is model-based in the sense that it is predicting actions and future state-action values, a numer of time steps into the future.
  It is model-free in that there is no explicit learning of the state transitions and rewards.
  It offers a new integration of planning and control.
- the method performs well on a set of 8 benchmark tasks, spanning 6 simple systems, a more challenging system (bipedwalker),
  and a more complex and realistic system building on visual input (CARLA autonomous driving).

Weaknesses:
- a better understanding of how this compares to model-based methods could be provided.  Like a model-based method, future plans can be generated and tested, although the model is implicit rather than explicit.  Experimental comparisons and discussions would help address this.  Regardless of the actual performance numbers, the results will be of interest to readers.

The form of the action generator is obviously an important choice.
From my understanding from Table 2, the recurrent models are not shared between actor & critic.
Is this a critical design choice?  Could both of those networks do well at reconstructing
the expected state sequence?  Would a take-home lesson be that it is more important to
predict next actions rather than next states?  And that working with entirely implicit next-state dynamics
(not even using a shared latent state for actor & critic) is simpler and more effective than leveraging
a learned latent dynamics model?  If the actor and critic could be conditioned on the real state
(imagine a setting with thea ability to do state resets, and not needing to count these "look-ahead" env steps)
would the learning improve?

Is the recurrent critic a key part of the solution, or could a more standard critic architecture also
be made to work?

What is the benefit of making the replan decision stochastic, as opposed to deterministic?

There is a large body of literature on the set of topics being discussed,
and so in many contexts, it could be useful to begin the list with "e.g.,"

Presumably there are many variations of the algorithm that could exist, with regard to
how many new plans are generated for each environment step; the choice of one new plan per env step is ultimately
just one arbitrary choice.

How important is the plan length hyperparameter?

Can you comment on the ability (or not) of the policy to generate multimodal futures?
Presumably not, given the action parameterization (unimodal multivariate Gaussians), and so does this
ever become a limitation?

Figure 2 could perhaps be labeled with "generative actor" and "generative critic".
It took me a while to understand that there was no other policy network, aside from the plan generator.

**Summary Of The Paper:**

A generative planning method is proposed, which sits somewhere between model-free and model-based methods.
No explicit model is learned. However, an action plan is generated using a recurrent action-plan generator,
paired with a similarly recurrent critic.  At each environment step, a new action plan is generated, and
the current action plan can be abandoned in favor of the new action plan if the estimated benefit is large enough.
The method builds on SAC.  Overall the benefits of the temporally extended action plans are:
(a) temporally-coordinated exploration; (b) more effective than action-repeat; (c) some degree of interpretability
given that an action-sequence plan represents then intent of a policy in a given state.

**Summary Of The Review:**

A novel method is presented, which, in the model-free setting, improves upon current alternatives for temporally extended actions,
in the mod, and model-free SAC. I believe that the results will be of strong interest to the community. The core idea is a creative one that performs well, and will promote future work and discussion.

---

> ### Author Response · Authors · 2021-11-23
> **Response to Reviewer 3VLM (Part 2/2)**
>
> **Q7: "What is the benefit of making the replan decision stochastic, as opposed to deterministic?"**
>
> This makes the replanning decision "soft" during rollout for training, providing more exploration opportunities when the Q values are not learned well in the early state of training.
>
> **Q8: "There is a large body of literature on the set of topics being discussed, and so in many contexts, it could be useful to begin the list with 'e.g.,' ..."**
>
> Thanks for the suggestions. We have revised our paper and added "e.g." before the citation list when appropriate. This indeed helped to improve the presentation.
>
> **Q9:  "Presumably there are many variations of the algorithm that could exist, with regard to how many new plans are generated for each environment step; the choice of one new plan per env step is ultimately just one arbitrary choice."**
>
> In this work, we have demonstrated the benefits of generating intentional action sequences in RL by using one simple instantiation. We agree with your comment that there are potential for other variations and further developments. We also listed this as one of our future work and hope this can serve as a starting point for other researchers in the community as well (Section 6).
>
> **Q10:  "How important is the plan length hyperparameter?"**
>
> We empirically observe that the performance is robust to this hyper-parameter for non-trivial length values. We provided some empirical results in Appendix A.7.
>
> **Q11: "Can you comment on the ability (or not) of the policy to generate multimodal futures ..."**
>
> The current model structure does not exclude the possibility of modeling multi-mode futures, as the samples from the unimodal multivariate Gaussian at the first time step are transformed by non-linear transformations. However, our training scheme does not have any particular processing for this and we did not focus on this particular property in the current work.
>
> **Q12: "Figure 2 could perhaps be labeled with "generative actor" and "generative critic" ..."**
>
> Thanks for the suggestion. We have added them to the caption of Figure 2 in the revised paper.
>
>
> &nbsp;
> &nbsp;
> &nbsp;
>
> **References**
>
>
> [1]  Danijar Hafner, et al. Dream to control: Learning behaviors by latent imagination. In International Conference on Learning Representations, 2020.
>
> [2] Deepak Pathak et al. Curiosity-driven exploration by self-supervised prediction. In International Conference on Machine Learning, 2017.
>
> [3] Williams, Ronald J., and David Zipser. "A learning algorithm for continually running fully recurrent neural networks." Neural computation 1.2 (1989): 270-280.

---

> > ### Comment · Reviewer_3VLM · 2021-11-27
> > **re: response**
> >
> > Thank you for the detailed response.  This covers all my questions.

---

> ### Author Response · Authors · 2021-11-23
> **Response to Reviewer 3VLM (Part 1/2)**
>
> We thank the reviewer for the positive review and insightful comments.
>
> Responses to the questions are below:
>
>
> **Q1: “a better understanding of how this compares to model-based methods could be provided ..."**
>
> Thanks for the suggestion. We have now added a paragraph on discussing the relationship with model-based RL in Section 4.
> We have also added a model-based RL baseline method Dreamer [1] according to your suggestion (Figure 3).
> In addition, we also added a mode-based reward bonus method (ICM) [2].
>
> **Q2: “the form of the action generator: the recurrent models are not shared between actor & critic. Is this a critical design choice?"**
>
> This design offers advantages such as the flexibility of structural design (e.g. the usage of a structural prior Eqn.(4) in plan generator), network capacity allocation (e.g., in cases where the modeling or computational requirements on policy and value are different) and learning (each is responsible for its own objective). (added discussion in Section 4).
> The current architecture is just one simple and reasonable example and we believe there is a potential to further improve the performance by improving over architectures. We leave the thorough exploration of this for future work (Section 6).
>
> **Q3: “the form of the action generator: could both of those networks do well at reconstructing the expected state sequence? Would a take-home lesson be that it is more important to predict next actions rather than next states"**
>
> We did not predict the next state explicitly. Predicting the next action is lightweight and is suitable in our case as our motivation is to improve action repeat sequences with a more flexible one using a light-weight planning module. Next state prediction may be helpful in some cases as it provides more supervision for the model training.
>
> **Q4: “the form of the action generator:  working with entirely implicit next-state dynamics (not even using a shared latent state for actor & critic ..."**
>
> This enables the usage of light-weight models for action sequence generation
> We have demonstrated that generating intentional action sequences for flat model-free RL algorithms is beneficial. The proposed approach is just one instantiation. We hope this work can be found useful to researchers that are interested in this problem and serve as an initial point for further improvements (Section 6).
>
> **Q5: “If the actor and critic could be conditioned on the real state (imagine a setting with the ability to do state resets, and not needing to count these "look-ahead" env steps) would the learning improve?"**
>
> This is a great point! If we understand your comment correctly, it is a form of teacher-forcing [3], which is a way to improve the training of recurrent models. For latent dynamics model, we could potentially also use teacher-forcing for training by conditioning on the real-state. However, there are also some challenges as state resetting might not always be available. We will further explore this in future work.
>
> **Q6: “Is the recurrent critic a key part of the solution, or could a more standard critic architecture also be made to work?"**
>
> One key requirement on the critic network is that it can generate a sequence of plan values, which are used in replanning. Any network architecture that satisfies this requirement can potentially be used (e.g. CausalConv or CausalTransformer). The current recurrent architecture for the critic is just one simple design.

---

### Official Review · Reviewer_ZK8i · 2021-11-02

**Correctness:** 3
**Technical Novelty And Significance:** 2
**Empirical Novelty And Significance:** 2
**Recommendation:** 6
**Confidence:** 4

**Main Review:**

# Strengths
This paper is well-written and tackles an important problem. The experiments clearly demonstrate that GPM improves exploration over naive baselines (e.g. greedy exploration), and the fact that the model does not need to be trained to explicitly model the dynamics is a nice advantage.

# Weaknesses
The main weaknesses of this paper are that it does not demonstrate strong empirical results, that the related works section is missing some important discussion, and that the method is missing some important details. I describe each concern in detail below.

## Method
Although the paper read well, the method section has some important details missing. For example, do the authors use n-step return in Equation 6? If so, how is n-determined? Also, where does P(s_t, a_t) and T(s, \tau) come from? Does the method assume access to the ground-truth dynamics model T? Also, how was l_commit_target chosen? How sensitive is the method to this hyperparameter, and is there evidence that it’s easier to choose l_commit_target than epsilon directly?


## Related work

The paper could greatly benefit from discussing the relationship to works such as [10-14], since these papers also implicitly learn models through model-free RL.

Another concern with the writing is that the claim that, “we provide a new aspect and opportunity for improving temporally consistent exploration from the perspective of planning” is a strong overstatement. There have been multiple sub-fields that have explored this idea of using planning to explore, including goal-conditioned reinforcement learning (RL) [1,2,3], hierarchical RL [4,5,6], and model-based RL [7,8,9], as well as many of the papers that the paper cites. I strongly suggest the authors reconsider the phrase of this claim and either remove that claim altogether, or be much more specific with the novelty that is being provided.

## Experiments

In terms of experiments, the paper does not compare to competitive exploration methods and the tasks, other than CARLA, are relatively easy. In particular, the paper would benefit from comparing to reward-bonus, option-based, goal-directed, and/or other planning exploration, or the paper would need to provide a strong justification for why comparing to these methods is unnecessary. Without these comparisons, I do not see evidence that, “we verify the effectiveness of the proposed method by comparing it with current state-of-the-art methods on a number of benchmark tasks”

Figures 4, 5, and 6 are a bit redundant: they all show that PGM have reasonable exploration and perform better than naive epsilon-greedy, but I think the space could be used more efficiently. For example, the paper would also be strengthened by spending more time describing the CARLA environment in the main paper.

The paper could also benefit from evaluating more challenging and diverse tasks. Pendulum, InvertedPendulum, InvertedDoublePendulum, and CartpoleSwingUp, are all similar and simple. I suggested the authors add comparisons on (e.g.) Atari domains or more challenging navigation tasks (BipedalWalker’s challenge is in learning _how_ to walk--not _where_ to walk).

## minor comments
 * Equation 1 does not describe the SAC update, which uses a soft update

## references

 - [1] Ren, Zhizhou, et al. "Exploration via Hindsight Goal Generation." Advances in Neural Information Processing Systems 32 (2019): 13485-13496.
 - [2] Pong, Vitchyr, et al. "Skew-Fit: State-Covering Self-Supervised Reinforcement Learning." International Conference on Machine Learning. PMLR, 2020.
 - [3] Pitis, Silviu, et al. "Maximum entropy gain exploration for long horizon multi-goal reinforcement learning." International Conference on Machine Learning. PMLR, 2020.
 - [4] Dayan, P. and Hinton, G. E. Feudal reinforcement learning. In Advances in Neural Information Processing Systems, 1993.
 - [5] Florensa, Carlos; Duan, Yan; Abbeel, Pieter.  Stochastic Neural Networks for Hierarchical Reinforcement Learning. In International Conference on Learning Representations (ICLR) 2017.
 - [6] Co-Reyes, John, et al. "Self-consistent trajectory autoencoder: Hierarchical reinforcement learning with trajectory embeddings." International Conference on Machine Learning. PMLR, 2018.
 - [7] Henaff, Mikael. "Explicit explore-exploit algorithms in continuous state spaces." NeurIPS. 2019.
 - [8] Shyam, Pranav, Wojciech Jaśkowski, and Faustino Gomez. "Model-based active exploration." International conference on machine learning. PMLR, 2019.
 - [9] Sekar, Ramanan, et al. "Planning to explore via self-supervised world models." International Conference on Machine Learning. PMLR, 2020.
 - [10] Oh, Junhyuk, Satinder Singh, and Honglak Lee. "Value Prediction Network." NIPS. 2017.
 - [11] Silver, David, et al. "The predictron: End-to-end learning and planning." International Conference on Machine Learning. PMLR, 2017.
 - [12] Schrittwieser, Julian, et al. "Mastering atari, go, chess and shogi by planning with a learned model." Nature 588.7839 (2020): 604-609.
 - [13] Weber, Théophane, et al. "Imagination-augmented agents for deep reinforcement learning." arXiv preprint arXiv:1707.06203 (2017).
 - [14] Srinivas, Aravind, et al. "Universal planning networks: Learning generalizable representations for visuomotor control." International Conference on Machine Learning. PMLR, 2018.

**Summary Of The Paper:**

This paper presents a method called generative planning method (GPM) to improve exploration in RL. GPM performs exploration by learning a planner (a map from state to a sequence of action, aka a “plan”)) and performing MPC with a special rule for whether or not to use the latest plan or keep the old plan. The planner is an auto-regressive model (specifically, a stochastic RNN) and is trained to maximize an auto-regressive Q-function. Each time step, the plan from the previous time step is shifted forward by one time step and compared to the newly generated plan. The plans are compared using their predicted Q-values, and the policy switches plans with some probability that increases monotonically with Q(new plan) - Q(old plan). The exact likelihood is determined by a hyperparameter, l_commit_target, that, intuitively, sets a soft target for how long a plan is typically kept. The authors compare GPM to a variety of action-repeat-based exploration methods, from epsilon greedy policies to policies with learned action repeat counts (DAR & TAAC) and find that it is competitive with or outperforms these methods on various low-dimensional robot domains, as well as the image-based CARLA environment. The authors also visualize the trajectories and qualitatively show that GPM improves exploration.

**Summary Of The Review:**

My main issue with the paper is that it claims to present a new exploration method, but it only compares to relatively simple exploration methods that only induce exploration via action repeat. The absence of more competitive baselines makes the significance of the paper relatively small. Moreover, some of the claims are overstated, some important methodological details are unclear, and the discussion of certain related works are missing.

After the author's response: The additional experiments and updated related work section have addressed my primary concerns. I strongly suggest adding the details about the method from the rebuttal (which I found useful) to the paper.

---

> ### Author Response · Authors · 2021-11-23
> **Response to Reviewer ZK8i (Part 2/2)**
>
>
>
> **Q3: “In terms of experiments, the paper does not compare to competitive exploration methods ..."**
>
> We are aware of the fact that is a large number of methods that could be regarded as related from some perspective. But due to practical constraints, we cannot enumerate them all but focus on the most relevant ones.
> We compared our proposed approach with several different action repeat based methods for several reasons:
> * they are closely related to GPM, both from the perspective of targeted problem (improving flat model-free RL methods), and from the algorithm structure (actor critic-like). In some sense, GPM can be viewed as generalization of action-repeat methods with more flexible action sequences;
>
> * the strong empirical results they have achieved. Action-repeat based exploration strategy has been proved to be very effective in literature and we have included some of the most recently developed approaches as our baselines from this category [e.g. EZ, DAR, TAAC].
>
> We follow your suggestion and added two more baselines (Figure 3):
> * ICM: which is relevant at lease from two points: 1) it improves on flat RL methods 2) a dynamics model is used (for computing the reward bonus);
> *Dreamer: which is a Model-based RL baseline method according to your suggestion.
>
> In general, reward-bonus, option-based, goal-directed are approaches complementary to the proposed method and can potentially be used together.
>
>
> **Q4: “The tasks, other than CARLA, are relatively easy ... The paper could also benefit from evaluating more challenging and diverse tasks...""**
>
> One reason we included some simple tasks  (e.g. pendulum) is that their low-dimensional state space is easier to be visualized (e.g. Figure 12).
>
> We also want to emphasize that CARLA based driving task is a more complicated image-based control task that is very challenging to RL algorithms, and GPM outperforms baselines methods on this very challenging task (Figure 4).
>
> We thank the reviewer for the suggestions on increasing the diversity of tasks, which could further improve the quality of the paper. Since CARLA driving is already a navigation task, in order to increase diversity, we have added an additional set of experiments on 4 Fetch manipulation tasks (Figure 3), making our evaluation set covering simple control, locomotion, navigation and manipulation.
>
> **Q5: “Equation 1 does not describe the SAC update, which uses a soft update"**
>
> Thanks for the comment. We have revised Equation (1) and related notations in the definition of Q below Equation (1).
>
> **Q6: “Experiments: Figures 4, 5, and 6 are a bit redundant"**
>
> Thanks for the comments. We have moved Figure 4 (of the original version) to appendix in our revised version, and used the space for more contents such as discussions on related works.
>
>
> &nbsp;
> &nbsp;
> &nbsp;
>
> **References**
>
> [EZ]  Will Dabney et al. Temporally-extended "-greedy exploration. In International Conference on Learning Representations, 2021.
>
> [DAR]: André Biedenkapp et al. TempoRL: Learning when to act. In International Conference on Machine Learning, 2021.
>
> [TAAC] Haonan Yu et al. TAAC: temporally abstract actor-critic for continuous control. Advances in Neural Information and Processing Systems, 2021.
>
> [ICM] Deepak Pathak et al. Curiosity-driven exploration by self-supervised prediction. In International Conference on Machine Learning, 2017.
>
> [Dreamer]  Danijar Hafner, et al. Dream to control: Learning behaviors by latent imagination. In International Conference on Learning Representations, 2020.

---

> > ### Comment · Reviewer_ZK8i · 2021-11-26
> > **Re: Rebuttal**
> >
> > Thank you for your response. The additional experiments and updated related work section have addressed my primary concerns. I strongly suggest adding the details about the method from the rebuttal (which I found useful) to the paper.
> >
> > I have increased my score to a 6.

---

> ### Author Response · Authors · 2021-11-23
> **Response to Reviewer ZK8i (Part 1/2)**
>
> We would like to thank the reviewer for the efforts in reviewing our work and the comments that helped to further improve the quality of the paper.
>
> Responses to the questions are below:
>
> **Q1: “Method: Although the paper read well ... do the authors use n-step return in Equation 6? If so, how is n-determined? Also, where does $P(s_t, a_t)$ and $T(s, \tau)$ come from? Does the method assume access to the ground-truth dynamics model $T$? Also, how was l_commit_target chosen? How sensitive is the method to this hyperparameter, and is there evidence that it’s easier to choose l_commit_target than epsilon directly?"**
>
> * We did not use n-step return in Eqn. (6). Q is trained by optimizing the objective presented in Eqn. (6) directly. Since we are learning Q value for an action sequence, this sum up of several steps of rewards directly follows our Q formulation (Q equation below Eqn.(5)). It can be treated as 1-step TD for multiple action steps. Note that our results is competitive to methods that use n-step return (e.g. DAR, TAAC).
>
> * $P(s_t, a_t)$ and $T(s, \tau)$ denotes the action-level and plan-level transition functions and we do not assume access to the ground-truth of them and we do not explicitly model the transitions as well. Instead, we use the experienced transitions as samples from them to estimate the gradients for learning the plan value function, similar to the practice done for typical TD-learning with point estimates.
>
>   Actually, this question is related to one of strengths of our method listed by you:
>
>   >"**Strengths**: … the fact that the model does not need to be trained to explicitly model the dynamics is a nice advantage."
>
> * l_commit_target can be regarded as the expected commitment length for the plans (e.g. for driving, how many seconds of execution of the current plan before switching on average), which is easier to interpret and therefore more intuitive to set. On the other hand, epsilon is a threshold for plan value differences, the range of which is unknown beforehand, therefore making the value setting for epsilon a more difficult task. Its setting was provided in Section A.6.1 along with other plan related details.
>
> * We investigated the sensitivity of the performance w.r.t. target commitment length in Section A 7.1 and found that the performance is not sensitive to this hyper-parameter and we set it as the half of the full length of the plan for experiments in the main paper.
>
>
> **Q2: “Related work: The paper could greatly benefit from discussing the relationship to works such as [10-14]… Another concern with the writing is that the claim… is a strong overstatement. There have been multiple sub-fields that have explored this idea of using planning to explore …"**
>
> We apologize for any confusion that might have been introduced in the original manuscript, but we want to clarify and address any confusions the reviewer might have.
>
> Our work is related to [10-12] in the sense that our plan value function is trained by predicting future values. However, these approaches do not leverage their implicit models for temporally extended exploration, which is the focus of our work, thus are orthogonal to our work. Also, our value function has a dedicated latent dynamics model that is not shared with the plan generator. This design offers advantages such as the flexibility of structural design and learning, as each is only responsible for its own objective.Therefore, the overall algorithmic structure resemble standard model-free RL methods, making the transition between them more natural.
>
> Imagination-augmented agent [13] uses the predictions from a model as additional contexts for a model-free policy. [14] embeds differentiable planning within a goal-directed policy and optimizes the action plan through gradient descent. Our work is different in that it uses a plan generator to directly generate action sequences conditioned on the current state, much like a model-free policy, without using additional context from a model-based prediction as in [13] or requiring expensive gradient based plan generation as in [14].
>
> The general idea of planning and plan-to-explore is broadly applicable and has appeared in many categories including hierarchical RL, goal-conditional RL as well as model-based RL. Differently, we focus on improving standard model-free RL methods by generating coordinated action sequence for temporally extended exploration, which has a different focus and complementary to these approaches.
>
> Actually, in the original submission, we have already included some discussions with them. However, as the discussion appeared in several places including introduction, background and related work, it might be less convenient for getting a clear picture.
>
> To further improve the presentation, we have revised the claim, re-organized related discussions and added some dedicated subsections for discussing the relationship of GPM with these related topics in Section 4 of the revised paper.

---

### Official Review · Reviewer_nNjd · 2021-11-08

**Correctness:** 3
**Technical Novelty And Significance:** 3
**Empirical Novelty And Significance:** 2
**Recommendation:** 8
**Confidence:** 3

**Main Review:**

Strengths
1. As far as I know, the idea is novel and very interesting. I really like the idea of connecting single-step action noise exploration and multi-step intentional exploration via planning under the perspective of planning for exploration.
2. The topic is definitely very related to the conference and of significant interest.
3. The paper is generally well-written and easy to follow. The pictorial demonstration is also very clear (fig.1 and fig.2).
4. The qualitative results (fig.4 & fig.5 & fig.6) are helpful to understand the exploration behaviors and are intuitive and interesting.

Weakness/questions
1. Testing environment. The tasks tested in the paper are pretty simple in my opinion (e.g. pendulum could be easily solved by a naive a2c agent). Considering the popular testing environment for continuous control are MuJoCo tasks, they should be tested. I am also wondering is there any particular reason why those environments are not considered in the first place? Furthermore, since this paper concerns exploration, it will also be useful to demonstrate the effectiveness in sparse-reward environments.
2. Presenting. I think there are some issues that could be improved to make the paper more clear. Some suggestions are as follows. 1) some important concepts are not explained well. For example, what does action-repeat mean, what is the dithering phenomenon... Those terms keep appearing in the paper and they should be explained more carefully. 2) terms being used. Throughout the paper, some (I think) similar terms are used differently and maybe interchangeably and I find it hard to understand. For example, temporally extended/coordinated/consistent/persistent...
3. for the plan generator, why are mu and sigma only generated for the current action? what's the reasoning behind this design choice?
4. for plan value function, why should be objective optimized in expectation to the length l? isn't that enough you optimize for L? As long as the plan has a high value, why do we care about high-value sub-plans?






**Summary Of The Paper:**

This paper proposes a method for exploration called Generative Planning method (GPM), which generates a multi-step action sequence such that the exploration is more temporally consistent and "intentional" compared to regular single-step action noise exploration. The multi-step action sequence is output by a generator with an RNN structure, and the generator is optimized by maximizing the plan value function. The authors show that their method GPM performs better than some other methods in several continuous control tasks and present some interesting qualitative results (e.g. state trajectory) showing that the exploration is more effective.

**Summary Of The Review:**

I believe the idea and the approach are quite novel and interesting, however, I do believe some extra experiments should be performed and some improvements over the paper presentation are needed.

---

> ### Author Response · Authors · 2021-11-23
> **Response to Reviewer nNjd**
>
> We thank the reviewer for the positive review and constructive comments.
>
> Responses to the questions are below:
>
> **Q1: “Testing environment. The tasks tested in the paper are pretty simple in my opinion..."**
>
> One reason we included some simple tasks  (e.g. pendulum) is that their low-dimensional state space is easier to be visualized (e.g. Figure 12). Mujoco locomotion tasks are not used as they do not benefit from action-repeat based methods according to previous work (TAAC), potentially because of their dense and not very delayed rewards (mostly x coordinate increment).
>
> We also want to emphasize that CARLA based driving task is a more complicated image-based control task that is very challenging to RL algorithms, and GPM outperforms baselines methods on this challenging task (Figure 4).  Since CARLA poses more challenges than other relatively simpler tasks, we've re-organized our experiment section to especially emphasize the weight of CARLA in our empirical evaluation.
>
> We thank the reviewer for the suggestions on sparse-reward based tasks, which could further improve the quality of the paper. We have added additional set of experiments on 4 more Mujoco-based sparse-reward tasks [1], where GPM performs competitively with other baseline methods (Figure 3).
>
> In addition, we also added two more baseline methods: a model based bonus-reward based exploration method ICM [2], and a model-based RL method Dreamer [3] according to your suggestion (Figure 3).
>
>
> **Q2: “Concepts such as action-repeat, dithering... and terms such as temporally extended/coordinated/consistent/persistent..."**
>
> Thanks for the great comments. These comments are valuable for improving the presentation of the work to reach a broader audience. We have revised our paper according to your suggestions (e.g. page 1 para 2 on explaining action-repeat, and we also unified the terms on exploration throughout the paper, while keeping a few where necessary with clear context).
>
> **Q3: “For the plan generator, why are mu and sigma only generated for the current action? what's the reasoning behind this design choice?"**
>
> This design choice facilitates the incorporation of prior on the temporal structure of action sequences. One example is give in Eqn.(4). In this case, if we initialize $g$ to generalize small values, then an action-repeat prior is incorporated. Basically, by incorporating noise only at the current step, we can have better control at the temporal consistency of the plan.
>
> **Q4: “For plan value function, why should be objective optimized in expectation to the length l? isn't that enough you optimize for L? As long as the plan has a high value, why do we care about high-value sub-plans?"**
>
> We train the plan value function for plans of different lengths to make it accurate not only for the full-length plans, but also for the shorter ones. The reason for this is that for replanning, we need to calculate the values of plans with a length that is shorter than the full length (denoted as $Q_{ l_{\rm old}}$), as detailed in Section 3.3.
> For plan generator training (Eqn. 5), it is used to ensure the sub-plans are also of high-quality.  Although this might not be necessary eventually, it helps with training, potentially due to reduced gradient variance.
>
>
> &nbsp;
> &nbsp;
>
> **References**
>
>
> [1] Matthias Plappert, et al. Multi-goal reinforcement learning: Challenging robotics environments and request for research. ArXiv, abs/1802.09464, 2018a.
>
> [2] Deepak Pathak et al. Curiosity-driven exploration by self-supervised prediction. In International Conference on Machine Learning, 2017.
>
> [3]  Danijar Hafner, et al. Dream to control: Learning behaviors by latent imagination. In International Conference on Learning Representations, 2020.

---

> > ### Comment · Reviewer_nNjd · 2021-11-29
> > **thanks you for the detailed and helpful replies**
> >
> > I would like to thank the authors for their detailed and helpful replies! They are clear and address most of my concerns. I am happy to increase my score to 7.

---

### Author Response · Authors · 2021-11-23
**Thanks for the reviews and  summary of key paper changes**

We thank all the reviewers (**R1**-nNjd, **R2**-ZK8i, **R3**-3VLM and **R4**-356g) for their reviews.
The reviewers appreciated the:
1. novelty of the method (**R1**-nNjd, **R3**-3VLM, **R4**-356g)
2. good performance and results (**R1**-nNjd, **R3**-3VLM, **R4**-356g)
3. interesting and interpretable plan (**R1**-nNjd, **R3**-3VLM, **R4**-356g)
4. good presentation (**R1**-nNjd, **R2**-ZK8i, **R3**-3VLM and **R4**-356g).

We summarize in the following the key changes made to the paper:
1. Clarified our first point of contributions with more clear explanations;
2. Added new subsections to Related Work for Hierarchical RL, Goal-conditioned RL and Model-based RL, including discussions on some of the references mentioned by the reviewers;
3. Added 4 sparse-reward-based manipulation tasks, further expanding the diversity of the evaluation tasks;
4. Added 2 more baselines: one model-based bonus reward method (ICM) and one model-based RL method (Dreamer);
5. Re-organized our experiment section to especially emphasize the weight of CARLA-task in our empirical evaluation due to the greater challenges it poses.

The major text changes in the revised paper are highlighted in blue for easier review.

---

### Decision · Program_Chairs · 2022-01-20

**Decision:**

Accept (Spotlight)

**Comment:**

This paper proposes an alternative approach to epsilon-greedy exploration by instead generating multi-step plans from an RNN, and then stochastically determining whether to continue with the plan or re-plan. The reviewers agreed that this idea is novel and interesting, that the paper is well-written, and that the evaluations are convincing, showing large improvements over epsilon-greedy exploration and more consistently strong performance than other baselines. While the original reviews contained some questions around discussion of related work and the simplicity of the evaluation environments, the reviewers felt these concerns were adequately addressed by the rebuttal. I agree the paper explores a very interesting idea and convincingly demonstrates its potential, and should be of wide interest to the deep RL community especially as it touches on many different subfields of RL: MBRL/planning, exploration, HRL, navigation, etc. I recommend acceptance as a spotlight presentation.